# Landslide susceptibility mapping on global scale using method of logistic regression

Le Lin[1,2]   Qigen Lin[1,2]   Ying Wang[1,2]

[1]Key Laboratory of Environmental Change and Natural Disaster of MOE, Beijing Normal University, No.19, XinJieKouWai St., HaiDian District, 100875, Beijing, China

[2]Academy of Disaster Reduction and Emergency Management, Beijing Normal University, No.19, XinJieKouWai St., HaiDian District, 100875, Beijing, China

*Correspondence to*: Ying Wang (wy@bnu.edu.cn)

**Abstract.** This paper proposes a statistical model for mapping global landslide susceptibility based on logistic regression. After investigating explanatory factors for landslides in the existing literature, five factors were selected to model landslide susceptibility: relative relief, extreme precipitation, lithology, ground motion and soil moisture. When building model, 70% of landslide and non-landslide points were randomly selected for logistic regression, and the others were used for model validation. For evaluating the accuracy of predictive models, this paper adopts several criteria including receiver operating characteristic (ROC) curve method. Logistic regression experiments found all five factors to be significant in explaining landslide occurrence on global scale. During the modeling process, percentage correct in confusion matrix of landslide classification was approximately 80% and the area under the curve (AUC) was nearly 0.87. During the validation process, the above statistics were about 81% and 0.88, respectively. Such result indicates that the model has strong robustness and stable performance. This model found that at a global scale, soil moisture can be dominant in the occurrence of landslides and topographic factor may be secondary.

**Keywords**

global scale; landslide susceptibility mapping; explanatory factors; logistic regression

## 1. Introduction

Landslides are a pervasive natural hazard, causing significant casualties and economic loss around the world (Budimir et al., 2015). Major news websites and online blogs from experts (such as The Landslide Blog, a thematic blog maintained by Prof. Dave Petley at the University of East Anglia) show that landslides almost occur every day. It is important and necessary to find out where the global landslide hotspot areas are and what factors can influence the occurrence of landslides. Such information would provide crucial reference for researchers and decision makers in some industries like insurance, and project managers in some non-governmental organizations (NGO). For those international and national insurance or reinsurance companies, such map will provide them with clear knowledge of landslide hotspots at a macro level, which will help them concentrate on those susceptible areas and make relevant marketing strategies like transferring risks (Bednarik et al., 2010). Geographers can also find it interesting for revealing spatial pattern of landslide distribution. To answer these questions, studies of global landslide susceptibility are required. Such research will help give a global perspective on landslides, which may encourage international cooperation for disaster risk reduction.

At present, research methods for landslide susceptibility mapping can be divided into three major

categories, qualitative factor overlay, statistical models and geotechnical process models (Dai and Lee, 2002). Generally, geotechnical process methods are developed from slope stability analyses and are applicable for site-specific landslides or when the ground conditions are quite uniform in the study area. Also, this method requires the landslide types to be known and relatively easy for analysis (Terlien et al., 1995; Wu and Sidle, 1995), and hence it is seldom used in large-scale landslide susceptibility mapping. In qualitative methods, landslide experts select landslide controlling factors and combine these factors into a susceptibility map, based on their knowledge and experience of landslide investigation. (Anbalagan, 1992; Pachauri and Pant, 1992). In contrast, statistical methods include statistical determination into combinations of explanatory factors (Carrara et al., 1991; Dhakal et al., 1999). Among these three types of methodologies, the latter two are widely applied in large-scale landslide susceptibility mapping. Relatively, reproducibility of results and subjectivity in landslide modelling can be the apparent disadvantages of the method of qualitative factor overlay. In recent time, large volumes of landslide inventories and multi-source data of landslide factors are gradually accessible to researchers and that make statistical methods are frequently used in landslide susceptibility mapping.

In statistical methods, Logistic regression model has been frequently used in geological hazard research and employed to explore the factors that influences landslides and determine landslide probability (Ayalew and Yamagishi, 2005; Van Den Eeckhaut et al., 2006). Compared with other statistical approaches, Brenning (2005) found that logistic regression models have a relatively low rate of error. Logistic regression can include dichotomous dependent variables (e.g. whether a landslide occurred) and independent variables, as well as categorical or continuous variables (Chang et al., 2007; Atkinson and Massari, 1998). The fact that landslide explanatory factors can be included in the model as either categorical or continuous variables gives logistic regression models a great advantage over multiple regression models, which can only include continuous variables. Finally, logistic regression models can be used to draw susceptibility maps when combined with GIS (Lee, 2005; Bai et al., 2010).

A landslide inventory provides the basis for quantitative zoning of landslide susceptibility. Location, date, type, size, causal factors and damage are supposed to be included in this database. A commonly used landslide inventory does not yet appear but some regional or national landslide databases are now well developed. In Europe, currently 22 out of 37 contacted countries have national landslide databases, and six other countries only have regional landslide databases. Those national databases contain about 633,700 landslides in total, of which about 75% are in Italy, and more than 10,000 landslides are in Austria, the Czech Republic, France, Norway, Poland, Slovakia, and the UK. In these 37 European countries, only six have sufficient information to perform risk analysis and one to perform a hazard analysis, while 14 countries can carry out at least a susceptibility analysis. Therefore, at a continental scale landslide zoning seems to be limited to landslide susceptibility modelling only. Restricted access to the data also make it difficult for these data to be applied in scientific research (Van Den Eeckhaut, and Hervás, 2012).

In the existing literature, there are few studies of landslide susceptibility that were carried out on a global scale; those that exist mainly used qualitative or semi-qualitative methodologies. For example, Mora and Vahrson (1994) proposed a method for assessing landslide susceptibility in tropical earthquake-prone areas that included three fundamental factors (slope, soil moisture, and lithology) and two triggering factors (extreme precipitation and ground motion). Nadim et al. (2006) applied the research of Mora and Vahrson (1994) to assess global landslide susceptibility and risk. Hong et al. (2007) selected six influencing factors (slope, elevation, soil type, soil texture, land cover type and drainage density) in the model of weighted linear combination (WLC). To obtain optimal weights combination, they tried

different combination of factor weights to make model results similar with the existing landslide susceptibility map of the USA. Finally, they drew a global landslide susceptibility map using the weights combination obtained above. Some scholars have also attempted to study global landslides with statistical methods. Farahmand and AghaKouchak (2013) used a global landslide inventory compiled by the National Aeronautics and Space Administration (NASA) to build a global landslide susceptibility model based on the method of Support Vector Machine (SVM) that includes three variables, satellite-sensed precipitation, digital elevation model (DEM) and land cover type. Compared with some complex numerical methods like SVM, logistic regression provides a simple method to produce global landslide susceptibility map, which would be helpful in disseminating this research and could encourage further model development for its simplicity in modeling. What's more, the result from logistic regression could illustrate the relative importance of different factors in explaining landslides, which could not be achieved by some numerical methods like SVM.

This paper addresses the gap in creating global landslide susceptibility maps using the widely used statistical method: logistic regression, and demonstrating the relative significance of different explanatory factors in global scale. In this paper, a global landslide inventory database is constructed and used for building a stepwise logistic regression model to evaluate global landslide susceptibility. Finally, a global landslide susceptibility map that visualizes this model is produced. In the landslide susceptibility model, five factors (extreme precipitation, soil moisture, lithology, relative relief and ground motion) are included as explanatory factors in stepwise logistic regression. In total, 70% of landslide and non-landslide events are randomly selected for logistic regression and the rest are used for model validation. It is found that such model has good explanatory power and performs well in model prediction. Landslide explanatory factors and the extent to which these factors influence landslide occurrence can be derived from model results directly without expert experience, which are rare in statistical assessment of global landslide susceptibility.

## 2. Explanatory factors

When assessing landslide susceptibility, the selection of explanatory factors is essential and significant. Typical explanatory factors from previous work (Table 1) fall into seven general categories, including topography, geology, hydrology, soil, precipitation, land cover and ground motion. Generally speaking, explanatory factors for landslides can be divided into fundamental factors and triggering factors (Nadim et al., 2006). Fundamental factors include environmental conditions that generate the potential of landslide occurrence, such as topography, lithology and soil. Triggering factors explain direct effects that drive slope instability, such as ground motion and extreme precipitation. In existing literatures, combination of trigger and susceptibility can influence landslide hazard level (Nadim et al., 2006). However, landslide model without landslide information like time and magnitude (like size, speed, kinetic energy or momentum of mass) cannot be correctly defined as hazard models (Guzzetti et al., 1999). Hence, in this paper, both fundamental factors and triggering factors are included to evaluate landslide susceptibility.

In existing studies of landslides at a regional scale, topography is regarded as a powerful explanatory factor for the occurrence of landslides (Dai and Lee, 2002; Lee and Min, 2001), and it is also demonstrated at a global scale (Hong et al., 2007). For most studies, topography includes relief characteristics such as elevation, slope gradient and slope aspect. At a global scale, factors such as elevation and slope gradient can be replaced by topographic index or relative relief, which indicate macroscopic differences in topography. Especially for landslide data with low location precision, using

factors such as elevation or slope gradient that precisely relate to landslide location will reduce the accuracy of landslide susceptibility analysis (Farahmand and AghaKouchak, 2013). Therefore, a general factor such as relative relief is more appropriate, and in this paper, relative relief is used to represent topography. Relative relief is defined as the difference between maximum and minimum elevation values within an area (Chauhan et al., 2010). Relative relief has been shown to be an important explanatory factor, and landslide occurrence is generally higher in high relative relief areas (Anbalagan, 1992).

For geology, attributes like rock age and rock type can be chosen, with data mainly coming from small regional geological surveys and field studies. Studies of global landslide susceptibility have shown that lithology is a fundamental factor (Nadim et al., 2006). Landslides are more likely to occur in some relatively later formed rocks with lower intensity and less likely in relatively earlier formed rocks with sufficient solidification and high intensity. Hence the factor of lithology is included in landslide model.

The water condition of the land surface also affects landslides. With the development of large data sharing frameworks for meteorological data, precipitation information is easily available and hence frequently used in landslide analysis (Farahmand and AghaKouchak, 2013). However, as Nadim et al. (2006) propose, soil moisture also can be proxy of water condition for it represents average moisture condition of the soil. Compared with mean annual precipitation, it can avoid the interruption of extreme precipitation, which can objectively reflect the possibility of slope instability in long period and can be taken as fundamental factor of landslide occurrence. Farahmand and AghaKouchak (2013) also recommend the use of soil moisture data in study of global landslide susceptibility. Therefore, soil moisture as an explanatory factor is adopted in this paper.

Ground motion and extreme precipitation are always analyzed as triggering factors of landslides, using data from field surveys and monitoring observations. Landslides are generally triggered by earthquakes or by heavy precipitation. Strong ground motion during earthquakes cause rocks to rupture, thus inducing landslides. As for rainfalls, rain and/or meltwater that reaches the ground surface infiltrates into the ground and forms groundwater. During this process, the pressure of the water that fills the void spaces between soil particles and rock fissures rises when the amount of water infiltrating into the ground increases. A rise in pore-water pressure causes a drop in effective stress, affecting the stability of a slope, and thus is a major cause of landslides and other sediment-related disasters (Matsuura et al., 2008). Intense rainfall is believed to be a cause of shallow landslides (Caine, 1980). Current studies of landslides consider ground motion and extreme precipitation as triggering factors (Umar et al., 2014; Nowicki et al., 2014; Nadim et al., 2006). Therefore, in this paper, ground motion and monthly extreme precipitation are used as triggering factors. In summary, this paper uses relative relief, soil moisture, lithology, monthly extreme precipitation and PGA as explanatory factors for global scale landslide susceptibility. The first three are fundamental factors, and the last two are triggering factors.

## 3. Methodology and Data
### 3.1 Study area
This paper considers global continental areas from 72 °N to 72 °S, excluding Greenland and the Antarctic continent. Because this research is specific to terrestrial landslides, oceans and areas covered by glaciers or ice sheets are excluded. The scope of this paper is also limited by data coverage for explanatory factors. As the coverage area of lithology is from 72 °N to 72 °S, therefore, the final susceptibility map is limited to such boundary.
### 3.2 Logistic regression model
What's more, Logistic regression models are commonly fitted in a stepwise manner (Budimir et al., 2015).

The general form of a logistic regression model is as follows:

$\text{logit(y)} = \beta_0 + \beta_1 x_1 + \beta_2 x_2 + \cdots + \beta_i x_i + e$         (1)

In Eq. 1, y is the dependent variable that reflects landslide occurrence, $x_i$ is the independent variable

related to explanatory factors, $\beta_0$ is a constant, $\beta_i$ is the regression coefficient for the explanatory factors,

and $e$ is the random error. The probability p of the dependent variable y can be expressed as follows in

Eq. 2:

$\text{p} = \dfrac{exp(\beta_0 + \beta_1 x_1 + \beta_2 x_2 + \cdots + \beta_i x_i)}{1 + exp(\beta_0 + \beta_1 x_1 + \beta_2 x_2 + \cdots + \beta_i x_i)}$         (2)

## 3.3 Independent variables

In this paper, explanatory factors are put into stepwise logistic regression model as independent variables.

All layer data of these explanatory factors are converted to the WGS 1984 geographical coordinate

system. Original resolution of factors is reserved as simple resampling cannot make real contribution to

the accuracy and precision of information provided in the layers.

Topographic data come from GTOPO30 (USGS, 2012), which is a global elevation dataset from Earth

Resources Observation and Science (EROS) Center. Its spatial resolution is 30 arc-seconds

(approximately 1 kilometer), and it covers the earth surface from 90 °N to 90 °S and 180 °E to 180 °W.

After obtaining the data, relative relief is calculated by a moving window method in ArcGIS with window

size of 0.5 arc-degree. From existing literatures, there is few statement about proper classification method

of relative relief. Relative relief is hence divided into 10 types with successive ordinal values from 1 to

10, using the natural breaks method of classification (Table 2).

Lithology data come from a geological map of the world at a 1:25,000,000 scale (the third version)

published by the Commission for the Geological Map of the World (CGMW, 2010) and UNESCO. In

the Mercator projection, the north and south boundaries of this map are set as 72 °N and 72 °S. As a

consequence, a large extent of the Antarctic continental coastline is visible, with a better delimitation of

the Southern Ocean. The southern half of Greenland is also visible (Bouysse, 2010). The lithology data

are rasterized with a spatial resolution of 0.01 °. Following Nadim et al. (2006), global lithology data can

be divided into 6 categories (Table 2). The spatial resolution of 0.01 ° was used because the primary

electronic map is vector-based. Its information can be greatly reserved by using small-scale raster when

converted into raster map, and a small-scale raster can fit the coastline well.

In this paper, the soil moisture index is used to represent the local soil humidity level. With data

products from the Center for Climatic Research at the University of Delaware, Willmott and Feddema

(1992) proposed a new soil moisture index. In this index, soil moisture was normalized to a range from

32   -1.0 to 1.0 with a spatial resolution of 0.5 °. Nadim et al. (2006) classified soil moisture data into levels

from 1 to 5 (Table 2), with higher values indicating greater humidity.

Monthly extreme precipitation with a repeat period of 100 years is calculated using historical

precipitation grid data over 50 years (from 1961 to 2010) from the GPCC Full Data Reanalysis

(Schneider et al., 2011). As no typical classification method for extreme precipitation exists in literatures,

this precipitation data is divided into 10 levels (Table 2) with a spatial resolution of 0.5 °, according to

the natural breaks classification method.

For ground motion, PGA with an exceedance probability of 10% over 50 years is included (that is, a

repeat period of 475 years). Data are from the global seismic hazard map created by the Global Seismic

Hazard Assessment Program (GSHAP) of the International Lithosphere Program (ILP). The map shows

PGA with an exceedance probability of 10% over 50 years and a spatial resolution of 0.1 °(Giardini et

al., 2003). Based on the methodology of Nadim et al. (2006), PGA can be divided into 10 levels (Table

2), with higher values denoting greater seismic hazard.

**3.4 Dependent variables**

The dependent variables that enter the model are global landslide inventory data and simulated non-landslide data.

This paper uses global landslide inventory data from a combined database. This database stores landslide information of two existing inventories: World Geological Hazard Inventory created by the Academy of Disaster Reduction and Emergency Management of Beijing Normal University (ADREM, BNU), and NASA global landslide inventory (refer to Kirschbaum et al. 2010 for details). The NASA global landslide inventory mainly collects landslides from several existing databases, including International Consortium on Landslides website (ICL; http://iclhq.org); International Landslide Centre, University of Durham (ILC; http://www.landslidecentre.org); The EM-DAT International Disaster Database (http://www.em-dat.net); International Federation of Red Cross and Red Crescent Societies field reports (http://www.ifrc.org); Reliefweb (http://reliefweb.int); humanitarian disaster information run by the United Nations Office for the Coordination of Humanitarian Affairs (OCHA); other online regional and national newspaper articles and media sources. The best resolution of the NASA global landslide inventory is 2 km. The items in World Geological Hazard Inventory were collected manually from news reports (e.g. mass media in China, Xinhua News, and Sina News) and records in books and journals (e.g. Galli and Guzzetti, 2007 and Gao, 1999). We searched information about landslide on Internet by using keywords like landslide and debris flow. Then we read these descriptions carefully to determine whether it is a landslide and locate it, and later put it into the database. Thus the main source of World Geological Hazard Inventory can be news data. By investigating these news, we can find out those landslides that are of large volume or of high danger, for these kinds of landslides can be of high news value. A large range of literatures, not only reviewed academic books and journals but also newspaper and local chronicles, was included to serve as the information sources so as to investigate those geological hazards which happened long time ago or in remote area. Such rich information sources can provide as more landslides as possible to reduce the uncertainty brought by limited landslide database. The best resolution of World Geological Hazard Inventory is 0.001 degree, about 100m. Two teams were assigned to develop and maintain this inventory. One team (about 10 persons) was responsible for collecting information from literatures and the other team (about 4 persons) was expected to check and review the items collected for data quality control. When combining these two databases, the occurrence of time provides crucial standard. When two landslide events have different time (month), they are both reserved in the new database. If two events have the same occurrence time (month) and their locations are close, investigation through details in source could determine whether they are from the same disaster. If yes, the record with higher spatial resolution is reserved and the one with lower resolution is dropped. Example of this inventory can be found in Table 3.

In the World Geological Hazard Inventory, the earliest event can be dated to 1618. In this database, there is 117 landslides occurred before 1975, 84 between 1975 to 2000, and 274 between 2000 and 2014. The landslide events in the NASA global landslide inventory mainly happened in 2003, 2007, 2008 and 2009. Hence these two databases are complementary and they can be emerged to produce a more complete landslide database. In all, the combined database stores landslide information like hazard type, occurrence time, location (including geographical coordinates and locating precision), fatalities and data sources. Currently, this database contains 2005 landslides, their location as showed in Fig. 1. This combined database includes landslides (debris slides, rotational slides, and slumps) and debris flows, following the landslide classification of Varnes (1984) and Cruden and Varnes (1996).

In order to demonstrate the representative of landslide data used in this research, the landslide overlay in Europe of this research is compared with the spatial distribution of landslides in the study of Van Den Eeckhaut et al. (2012). As showed in Fig. 2, it is found that the spatial overlay of landslide samples in the research of European landslide susceptibility modelling is quite similar with that of the combined landslide database in this research. It is estimated that there is about 60% agreement between these two landslide distributions in general. The landslides in Europe mainly distribute in mountainous areas like the Alps and the Balkan.

Non-landslide events come from generating random points. Because landslide location accuracy is approximately 0.25 °, a buffer zone is created around the existing landslide points with a radius of 0.25 ° to represent the location range of each landslide event. The buffer zone is then removed from the global continent area and the other part on global continent forms potential non-landslide area. The quantity of non-landslide points should be carefully considered. Most studies use an equal amount of landslide points and non-landslide points (Dai and Lee, 2002; Kawabata and Bandibas, 2009; Chau and Chan, 2005; Costanzo et al., 2014; Regmi et al., 2014; Mathew et al., 2009). However, a few authors prefer an unequal amount (Van Den Eeckhaut et al., 2012; Felicisimo et al., 2013). For example, Van Den Eeckhaut et al. (2006) use 5 times as many non-landslide cells as landslide cells, and Farahmand and AghaKouchak (2013) use 10 times as many non-landslide cells as landslide cells. In order to make sensitivity test on the landslide susceptibility model in the paper and also reduce the uncertainty included by random non-landslide, 5 non-landslide sets which each had equal number as landslides were created using random sampling without replacement. To validate the landslide model, method of splitting datasets is applied (Van Den Eeckhaut et al., 2012). For each dataset, 70% of landslide and non-landslide are randomly selected for modeling, and the remaining 30% are used for validation.

Confusion matrix and Akaike's information criterion value (AIC) (Allison, 2001; Van Den Eeckhaut et al., 2006) are applied to assess model performance. In addition, this paper also adopts a receiver operating characteristic (ROC) curve to evaluate model effectiveness. The ROC curve helps to validate a model graphically (Swets, 1988), providing an analysis based on true-positive and false-positive rates. With higher area under this curve (AUC), such model is demonstrated to perform well in prediction (Mathew et al., 2009).

## 4. Results

The results and validation of the logistic regression models for 5 datasets are shown in Table 4. It is found that among these 5 datasets, percentage correct in confusion matrix ranges from 78.7% to 80.4% during the modeling process and from 79.9% to 82.1% during the validation process. Generally, the logistic regression models in this study show high accuracy in confusion matrix. For the 5 datasets, their AUC values range from 0.8685 to 0.8846 when modeling (Fig. 3) and from 0.8809 to 0.8933 when validating (Fig. 4). On average, the AUC value in the logistic regression model is approximately 0.88, which indicates a relatively great performance in prediction.

By using the principle of high percentage correct in confusion matrix, high AUC value and low AIC value, the regression model from dataset 2 was selected as the global landslide susceptibility model. This model is then used to analyze the importance of the explanatory factors on landslides and employed in landslide susceptibility mapping. The formula of the best model is as follows:

$$\begin{cases} P = \frac{f}{1+f} \\ f = Exp(-5.7047 + 0.5528 * S + 0.1958 * A + 0.1245 * L + 0.3159 * R + 0.2957 * E) \end{cases} \quad (3)$$

where P stands for the probability of landslides, and S, A, L, R, and E stand for landslide explanatory factors of soil moisture, PGA, lithology, relative relief and extreme precipitation, respectively.

In the model above, all variables are significant at the 1% confidence level. The coefficients of each factor show that the greatest contribution to landslide occurrence comes from soil moisture, which has a coefficient of approximately 0.6. The next most important factors are relative relief and extreme precipitation, with coefficients of approximately 0.3. The contribution of PGA and lithology is relatively low, with a coefficient of approximately 0.2 and 0.1.

A table with the number of landslides in each continent in global inventory and in each data set used to model and validate is displayed, which will help readers understand how spatial representative the data sets used are (Table 5). It can be found that there is a small amount of landslide records in Africa. However, when either in the modelling process or validation process, different amount of landslides and non-landslides in African was selected. From Fig. 3 and Fig. 4, it is demonstrated that the results from every five datasets are relatively stable and high, which means the model built can be applied effectively in Africa. Otherwise, the results of five datasets may be different.

A global landslide susceptibility map can be drawn using the model in Eq. 3. Based on existing susceptibility classification methods from Guzzetti et al. (2006), Van den Eeckhaut et al. (2012), this map classifies susceptibility levels according to breakpoints of 0.4, 0.6, 0.7 and 0.9. These breakpoints define a susceptibility map with 5 levels, i.e. very low, low, moderate, high, very high (Fig. 5).

The susceptibility map shows that global landslide hotspots are the Alps, the Iranian Plateau, the Pamirs, the southern Qinghai-Tibet Plateau, the mountainous region of southwestern China, the islands in the western Pacific Ocean, including Japan, the Philippines, Malaysia, Indonesia and New Zealand, northeastern North America, Central America and the Andes in South America.

## 5. Discussion

To evaluate the accuracy of susceptibility map produced in this research, the global landslide susceptibility map is compared with four studies from the current literature that focus on large-scale landslide susceptibility. In regional scale, two landslide susceptibility maps, i.e. European (Van Den Eeckhaut et al. 2012) and Chinese (Liu et al. 2013), are selected. In global scale, the studies of Nadim et al. (2006) and Hong et al. (2007) are selected.

Comparing the European landslide susceptibility map drawn by Van Den Eeckhaut et al. (2012) with the European part of susceptibility map in this study (Fig. 6 (a)), similar areas of high landslide susceptibility can be observed. The former map includes two levels (denoted High and Very High) as high susceptibility with a landslide probability of over 0.8, and this study also includes two levels (Levels 4 and 5) as high susceptibility with a probability over 0.7. The two maps have similar high susceptibility areas. Thus, for Europe, landslide susceptibility map in this study agrees with existing related study.

Comparing the Chinese landslide susceptibility map drawn by Liu et al. (2013) with the China part of susceptibility map in this study (Fig. 6 (b)), the former map includes two levels (Levels 4 and 5) as susceptible with a landslide probability of over 0.6. Map in this study includes three levels (denoted Levels 3, 4 and 5) as susceptible with landslide probability of over 0.6. The main differences between the two maps are in the western Sichuan Basin and southern Tibet, which is famous for its high elevation and intense relative relief. This study applies many landslide cases in these areas. However, in the landslide database of Liu et al. (2013), only a few landslides occur in these areas. This discrepancy is the reason for the differences between the two maps.

As for landslide susceptibility at global scale, Nadim et al. (2006) and Hong et al. (2007) have ever

made magnificent efforts on such topic. One global landslide susceptibility map (please refer to Fig. 7 in Nadim et al. (2006)) has five levels (Levels 5, 6, 7, 8 and 9) as susceptible, while the map from this study includes three levels (Levels 3, 4 and 5) as susceptible. In general, the susceptible areas of these two maps are fairly similar except in Madagascar and the eastern Indo-China Peninsula.

Another global landslide susceptibility map (please refer to Fig. 3(a) in Hong et al. (2007)) has two levels (Levels 4 and 5) as susceptible, compared to map in this study, which has three levels (Levels 3, 4 and 5) as susceptible. These two maps are similar over Asia, Europe and Africa. However, it is noted that map of Hong et al. (2007) also differs from map of this study in that it shows high landslide susceptibility in central and southern India, and low landslide susceptibility in equatorial islands such as Malaysia, Indonesia, and the Philippines. We believe that the classification of landslide susceptibility of this research can be more scientific and closer to the existing conditions.

With the development of global DEM products, DEM with finer resolution is now available to the public. The NASA Shuttle Radar Topographic Mission (Jarvis et al., 2012) has provided digital elevation data for over 80% of the globe. This data is currently distributed free of charge. The SRTM data is available as 3 arc second (approx. 90m resolution) DEM covering the globe from 60 °N to 60 °S. The 1-arc-second data product was also produced and now is available for all countries. To explore the sensitivity of DEM on model result, experiments have also been performed when following all the procedures stated above, but using SRTM 90m DEM as source of topography. As showed in Table 6, landslide susceptibility model with 90m DEM had no significant difference (only an increase about 0.005 in AUC) with those models using 1km DEM (AUC in modelling, from 0.8768 to 0.8818; AUC in validation, from 0.8871 to 0.8929). When location precision of landslide is not that good, using finer DEM cannot help to increase the accuracy of landslide susceptibility analysis. DEM with coarser resolution (i.e. 1km DEM) is recommended as the topographical factor in global landslide susceptibility mapping.

The accuracy of logistic regression model in this paper is quite high compared with that of similar experiment that is performed at national scale (Lin et al., 2017) or local scale (Wang et al., 2016), which really exceeds expectation. To have one single model to explain the occurrence of past landslides events in global scale may be difficult, but the result of model in this paper shows that the factors and their weights in this research can actually provide good explanation of global landslide occurrence in one model.

For the incompleteness of landslide inventory in the global geological hazard database of this study, the landslides included in this study may represent only a subset of the total landslides around the world. Studying global landslide susceptibility in a more comprehensive and objective way requires a more complete global landslide inventory. As for the weights of factor, actually they cannot provide adequate accuracy when building landslide model in local scale. However, we have determined to compare this research with those performed in local scale to investigate the rules of landslide occurrence in different scales in the coming future.

The main focus of this research is global landslide susceptibility assessment, and hence the landslides in database of this research should be representative on global scale, i.e. having large volume or causing significant loss. The landslides in our database can meet such requirement and are either of large magnitudes or causing severe life loss or economic loss, which are hence easily reported by news agencies. The global landslide susceptibility map built on this database can inevitably underestimate the landslide susceptibility in some sparsely populated areas or less developed areas. However, if we don't following the guidelines, in our database there will be a large numbers of landslides that are occurred in

countries with good landslide catalogue and few in countries with poor landslide catalogue. Such model may lead to overestimation of landslide susceptibility in countries with rich landslide records and underestimation of landslide susceptibility in countries with poor landslide records. This may not be good for improving the accuracy of the map of global landslide susceptibility. Hence we think that the landslide database in our research is relatively high in representative and reliable. We will explore the use of big data on Internet in building more comprehensive landslide database in our next research and tries to enhance the studies of landslide susceptibility when landslide catalogues from various countries can be easily accessed in the future.

## 6. Conclusions

This paper applies stepwise logistic regression model to study landslide susceptibility on global scale. After investigating the explanatory factors for landslides in the existing literature, five explanatory factors: extreme precipitation, lithology, relative relief, ground motion and soil moisture, are selected. These factors are used to build a landslide susceptibility model through stepwise logistic regression based on landslides recorded in a combined global landslide inventory. It is found that the five explanatory factors perform well in explaining the occurrence of landslides on a global scale. Percentage correct in confusion matrix of landslide classification during modeling ranges from 78.7% to 80.4%, with an AUC value from 0.8685 to 0.8846. During validation, percentage correct in confusion matrix ranges from 79.9% to 82.1%, with an AUC value from 0.8809 to 0.8933. The results from those datasets are similar, and the coefficients and ranks of each explanatory factor are relatively stable, which suggests that the model is both robust and accurate.

Existing studies of landslide susceptibility generally use topography as an explanatory factor (Budimir et al., 2015). However, on a global scale, topography is not always the primary factor for landslide occurrence. For example, Hong et al. (2007) gives priority to slope when building their global landslide model, and friction has the highest regression coefficient in model for earthquake-induced landslides (Nowicki et al., 2014). The present study shows that on global scale, soil moisture is the most important factor, while topography (relative relief in this study) is secondary. Additionally, this study shows that soil moisture has significantly explanatory power for landslide occurrence on a global scale. Therefore, it may suggest that future work of landslide susceptibility should consider the influence of soil water condition and long-term precipitation when studying global landslide susceptibility.

**Acknowledgments**

This work was supported primarily by the National Natural Science Funds of China (41271544), and National Key Technology R & D Program of the Twelfth Five-Year Plan of China (No. 2012BAK10B03).

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

1  **Table 1** Brief summary of explanatory factors in landslide susceptibility assessment for regional scale

2  and global scale

| Factors | Geographic scale of study | |
|---|---|---|
| | Regional | Global |
| topography | slope gradient, slope aspect, elevation, plan curvatures, profile curvatures (1)*, slope morphology (2), standard deviation of slope gradient (6) | Median, minimum, and maximum slope values from DEM (10), topography index (11), slope angle (12), elevation (13) |
| geology | lithology (1), density of geological boundaries (3), distance to geological boundaries (3), weathering depth (4), tectonic uplift (9), geological age (6) | lithology (12) |
| hydrology | proximity to drainage lines (2), water conditions (4), drainage density (5), distance from river, stream power index (SPI) (7) | drainage density (13) |
| soil | texture, material, soil thickness (5), topographic wetness index (TWI)(7), soil type, soil moisture (6) | material strength (10), soil wetness (10), soil moisture (12), soil type (13), soil texture (13) |
| precipitation | rainfall (7), total monthly precipitation (8), annual precipitation (9) | precipitation rates from rainfall accumulations in the past (11), extreme monthly rainfall with 100 years return period (12) |
| land cover | vegetation cover (2,4), age, diameter and density of timber for vegetation (5), land use/cover (7), road construction (8) | land use and land cover (11,13) |
| ground motion | peak ground acceleration (7), earthquake and seismic shaking (8) | peak ground acceleration and peak ground velocity (10) |

*Numbers in the table indicate related studies, they are: (1) Ayalew et al., 2004; (2)Dai and Lee, 2002; (3)Kawabata and Bandibas, 2009; (4) Ercanoglu and Gokceoglu, 2002; (5)Lee and Min, 2001; (6)Van Den Eeckhaut et al., 2012; (7)Umar et al., 2014; (8) Alimohammadlou et al., 2014; (9) Erener and Duzgun, 2010; (10) Nowicki et al., 2014; (11) Farahmand and AghaKouchak, 2013; (12) Nadim et al., 2006; (13) Hong et al., 2007.

1 **Table 2** Input variables used in logistic regression analysis

| Dependent variables: landslide location | Data provider | Map details |
| --- | --- | --- |
| World Geological Hazard Inventory | ADREM, BNU | Point |
| Global landslide inventory | NASA | Point |

| Independent variables | Sources | Map details |
| --- | --- | --- |
| **Relative relief** (unit: m) | | |
| Classification method: natural breaks | GTOPO and SRTM DEM | 30/3 arc-second |
| (1. <=80; 2. 80-264; 3. 264-520; 4. 520-844; 5. 844-1226; 6. 1226-1672; 7. 1672-2232; 8. 2232-2982; 9. 2982-4024; 10. >4024) | | |
| **Lithology** | | |
| Classification method: refer to Nadim et al. (2006) | Geological map of the world | Polygon |
| (0. Undifferentiated facies, Ophiolitic complex, Endogenous rocks, | at a 1:25,000,000 by | (rasterized into |
| Oceanic crust; 1. Extrusive volcanic rocks: Precambrian, Proterozoic, | Commission for the | 0.01 arc-second) |
| Paleozoic and Archean, Endogenous rocks (plutonic and/or | Geological Map of the World | |
| metamorphic): Precambrian, Proterozoic, Paleozoic and Archean; 2. | (CGMW) and UNESCO | |
| Old sedimentary rocks: Precambrian, Archean, Proterozoic, Paleozoic, | | |
| Extrusive volcanic rocks: Paleozoic, Mesozoic, Endogenous rocks: | | |
| Paleozoic, Mesozoic, Triassic, Jurassic, Cretaceous; 3. Sedimentary | | |
| rocks: Paleozoic, Mesozoic, Triassic, Jurassic, Cretaceous, Extrusive | | |
| volcanic rocks: Mesozoic, Triassic, Jurassic, Cretaceous, Endogenous | | |
| rocks: Meso-Cenozoic, Cenozoic; 4. Sedimentary rocks: Cenozoic, | | |
| Quaternary, Extrusive volcanic rocks: Meso-Cenozoic; 5. Extrusive | | |
| volcanic rocks: Cenozoic) | | |
| **Soil moisture index** | | |
| Classification method: refer to Nadim et al. (2006) | Willmott and Feddema (1992) | 0.5 arc-second |
| (1. -1.0 ~ -0.6; 2. -0.6 ~ -0.2; 3. -0.2 ~ +0.2; 4. +0.2 ~ +0.6; 5. +0.6 ~ +1.0) | | |
| **Monthly extreme rainfall with return period of 100 years** (unit: mm) | | |
| Classification method: natural breaks | calculated using historical | 0.5 arc-second |
| (1. <=55; 2. 55-150; 3. 150-250; 4. 250-365; 5. 365-500; 6. 500-650; 7. 650-850; 8. 850-1100; 9. 1100-1650; 10. >1650) | precipitation grid data over 50 years (from 1961 to 2010) from the GPCC Full Data Reanalysis | |
| **PGA with an exceedance probability of 10% over 50 years** (unit: m*s$^{-2}$) | | |
| Classification method: refer to Nadim et al. (2006) | Global seismic hazard map | 0.1 arc-second |
| (1. 0.00-0.50; 2. 0.51-1.00; 3. 1.01-1.50; 4. 1.51-2.00; 5. 2.01-2.50; 6. 2.51-3.00; 7. 3.01-3.50; 8. 3.51-4.00; 9. 4.01-4.50; 10. >4.50) | created by the Global Seismic Hazard Assessment Program (GSHAP) of the International Lithosphere Program (ILP) | |

1    **Table 3** Example of landslide inventory in World Geological Hazard Inventory created by ADREM,

2    BNU

| ID | Hazard type | Date | Country | Continent | Location | Longitude/Latitude | Death | Lost | Injured | Location precision (°) | Sources |
|---|---|---|---|---|---|---|---|---|---|---|---|
| …… | | | | | | | | | | | |
| 000159 | Debris flow | 2005.6.1 | U. S. | North America | Laguna Beach, Los Angeles, California | 33°32′32.63″N 117°46′18.10″W | 0 | 0 | 2 | 0.05 | Sina News |
| …… | | | | | | | | | | | |
| 000168 | Landslide | 2010.11.4 | Costa Rica | South America | San Antonio, San José | 9°55′37.48″N 84°04′55.24″W | 20 | 12 | 0 | 0.1 | Xinhua News |
| …… | | | | | | | | | | | |
| 000403 | Debris flow | 2010.8.7 | China | Asia | Zhouqu, Gansu | 33°47′10.56″N 104°22′7.24″E | 1463 | 302 | 2244 | 0.1 | Xinhua News |
| …… | | | | | | | | | | | |
| 000465 | Landslide | 2008.6.29 | Côte d'Ivoire | Africa | Abidjan | 5°20′10.74″N 4°1′39.90″W | 7 | 0 | 4 | 0.01 | Sina News |
| …… | | | | | | | | | | | |

1  **Table 4** Model results of stepwise logistic regression for each dataset

| Dataset | Intercept | Soil moisture | PGA | Lithology | Relative relief | Extreme precipitation | AIC[†] | Modeling | | Validation | |
| --- | --- | --- | --- | --- | --- | --- | --- | --- | --- | --- | --- |
| | | | | | | | | Percentage correct | AUC | Percentage correct | AUC |
| Set 1 | -5.7898[***] | 0.5567[***] | 0.1196[***] | 0.1885[***] | 0.3583[***] | 0.2798[***] | 2511.2 | 0.801 | 0.8755 | 0.810 | 0.8914 |
| **Set 2** | **-5.7047[***]** | **0.5528[***]** | **0.1958[***]** | **0.1245[**]** | **0.3159[***]** | **0.2957[***]** | **2468.4** | **0.797** | **0.8789** | **0.821** | **0.8933** |
| Set 3 | -5.9134[***] | 0.5980[***] | 0.1803[***] | 0.1583[***] | 0.3312[***] | 0.2924[***] | 2421.8 | 0.804 | 0.8846 | 0.799 | 0.8812 |
| Set 4 | -5.6525[***] | 0.5432[***] | 0.1704[***] | 0.1073[**] | 0.3344[***] | 0.2977[***] | 2483.8 | 0.798 | 0.8766 | 0.804 | 0.8809 |
| Set 5 | -5.3490[***] | 0.5426[***] | 0.1663[***] | 0.1100[**] | 0.3022[***] | 0.2625[***] | 2564.5 | 0.787 | 0.8685 | 0.814 | 0.8886 |
| Average | --- | --- | --- | --- | --- | --- | 2489.9 | 0.797 | 0.8768 | 0.810 | 0.8871 |

[†]These statistics of AIC are based on the model with intercept and covariates

[**]Coefficients are significant at 1% confidential level

[***] Coefficients are significant at 0.1% confidential level

1    **Table 5** Numbers of landslides and non-landslides in each dataset.

| Continent | Landslides | Dataset 1 | | Dataset 2 | | Dataset 3 | | Dataset 4 | | Dataset 5 | |
|---|---|---|---|---|---|---|---|---|---|---|---|
| | | Modelling (70%) | Validation (30%) | Modelling (70%) | Validation (30%) | Modelling (70%) | Validation (30%) | Modelling (70%) | Validation (30%) | Modelling (70%) | Validation (30%) |
| Asia | 1205 | 847:348 | 358:163 | 848:394 | 357:157 | 838:383 | 367:155 | 849:364 | 356:162 | 847:393 | 358:165 |
| Africa | 69 | 55:317 | 14:129 | 50:307 | 19:114 | 47:324 | 22:130 | 47:315 | 22:129 | 49:302 | 20:126 |
| Europe | 121 | 94:211 | 27:70 | 86:212 | 35:116 | 87:200 | 34:91 | 88:206 | 33:88 | 85:251 | 36:106 |
| North America | 425 | 274:235 | 151:98 | 296:235 | 129:98 | 298:226 | 127:107 | 286:230 | 139:110 | 286:195 | 139:83 |
| South America | 133 | 93:189 | 40:98 | 86:174 | 47:75 | 99:179 | 34:78 | 97:193 | 36:80 | 96:144 | 37:64 |
| Oceania | 52 | 40:103 | 12:44 | 37:81 | 15:42 | 34:91 | 18:41 | 36:95 | 16:33 | 40:118 | 12:58 |
| Total | 2005 | 1403:1403 | 602:602 | 1403:1403 | 602:602 | 1403:1403 | 602:602 | 1403:1403 | 602:602 | 1403:1403 | 602:602 |

Numbers in left represent numbers of landslides, numbers in right represents numbers of non-landslides.

1    **Table 6** Results of model based on global SRTM DEM (90m)

| Dataset | Intercept | Soil humidity | PGA | Lithology | Relative relief | Extreme precipitation | AIC† | Modeling | | Validation | |
| --- | --- | --- | --- | --- | --- | --- | --- | --- | --- | --- | --- |
| | | | | | | | | Percentage correct | AUC | Percentage correct | AUC |
| Set 1 | -5.8362*** | 0.5359*** | 0.1173*** | 0.1876*** | 0.3585*** | 0.2809*** | 2495.9 | 0.802 | 0.8767 | 0.818 | 0.8939 |
| Set 2 | -5.7546*** | 0.5441*** | 0.1980*** | 0.1222** | 0.3151*** | 0.2947*** | 2436.6 | 0.799 | 0.8826 | 0.828 | 0.8980 |
| Set 3 | -5.9650*** | 0.5850*** | 0.1791*** | 0.1604*** | 0.3328*** | 0.2919*** | 2384.0 | 0.815 | 0.8888 | 0.810 | 0.8861 |
| Set 4 | -5.7457*** | 0.5503*** | 0.1682*** | 0.1060** | 0.3393*** | 0.2925*** | 2437.6 | 0.808 | 0.8822 | 0.806 | 0.8856 |
| Set 5 | -5.5849*** | 0.5618*** | 0.1629*** | 0.1236** | 0.3093*** | 0.2745*** | 2479.4 | 0.799 | 0.8785 | 0.815 | 0.9008 |
| Average | --- | --- | --- | --- | --- | --- | 2446.7 | 0.805 | 0.8818 | 0.815 | 0.8929 |

†These statistics of AIC are based on the model with intercept and covariates

**Coefficients are significant at 1% confidential level

***Coefficients are significant at 0.1% confidential level

1    **Fig.1** Landslide location in the combined landslide database

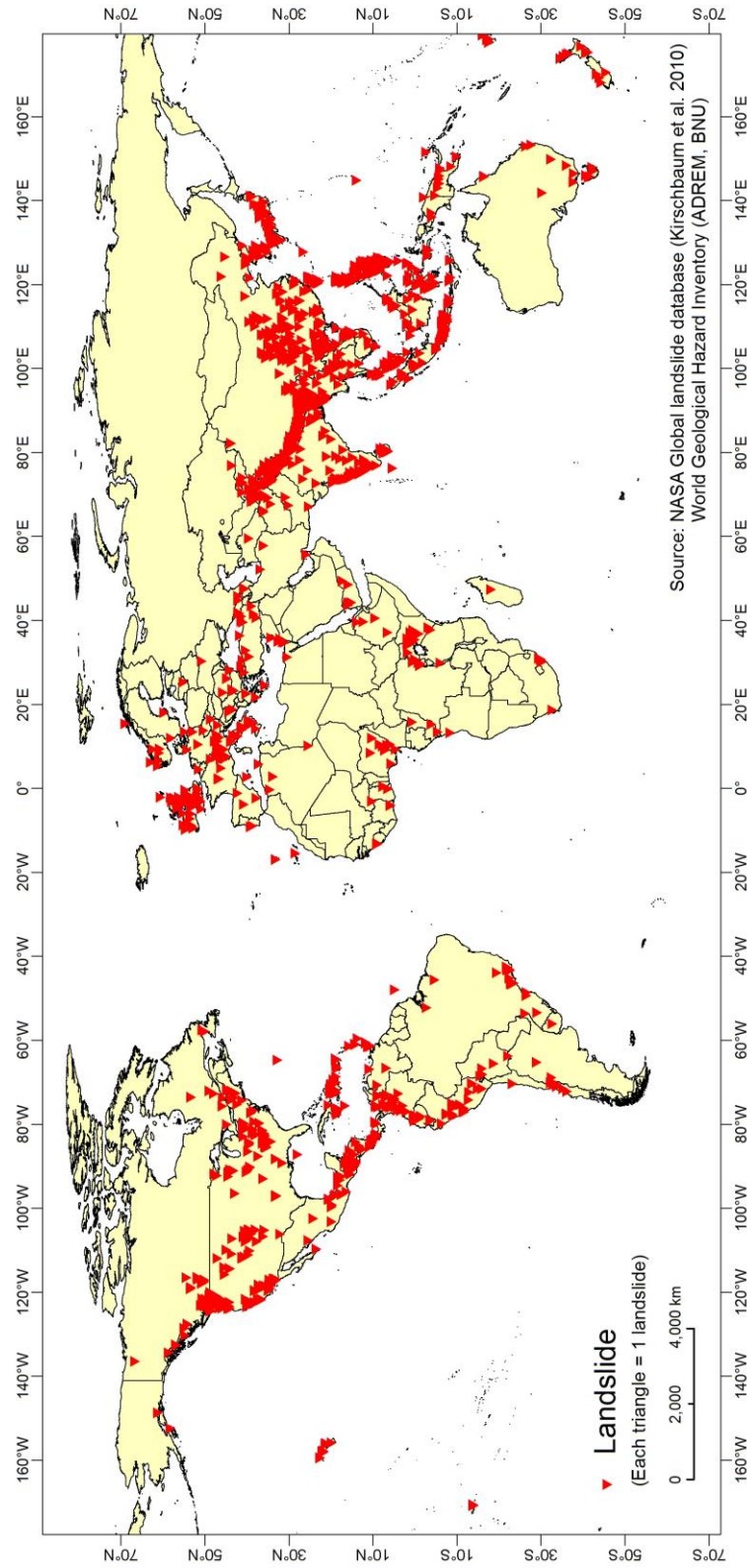

1    **Fig.2** Comparison of landslide overlay in Europe

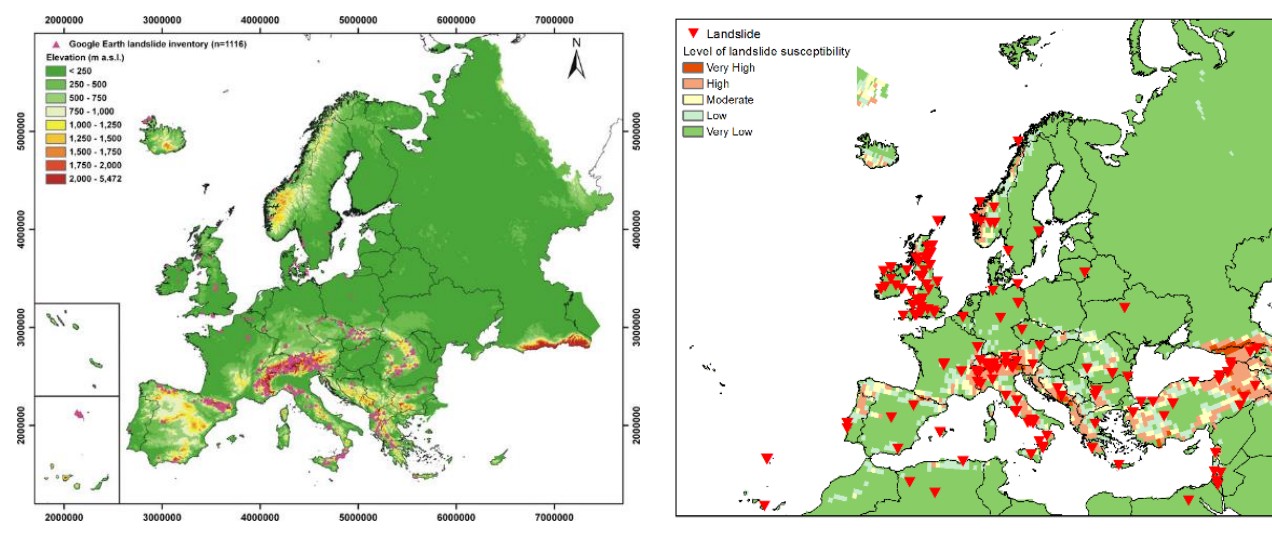

(a) from Van Den Eeckhaut et al. (2012)                    (b) from this research

1      **Fig.3** ROC curve of modeling process

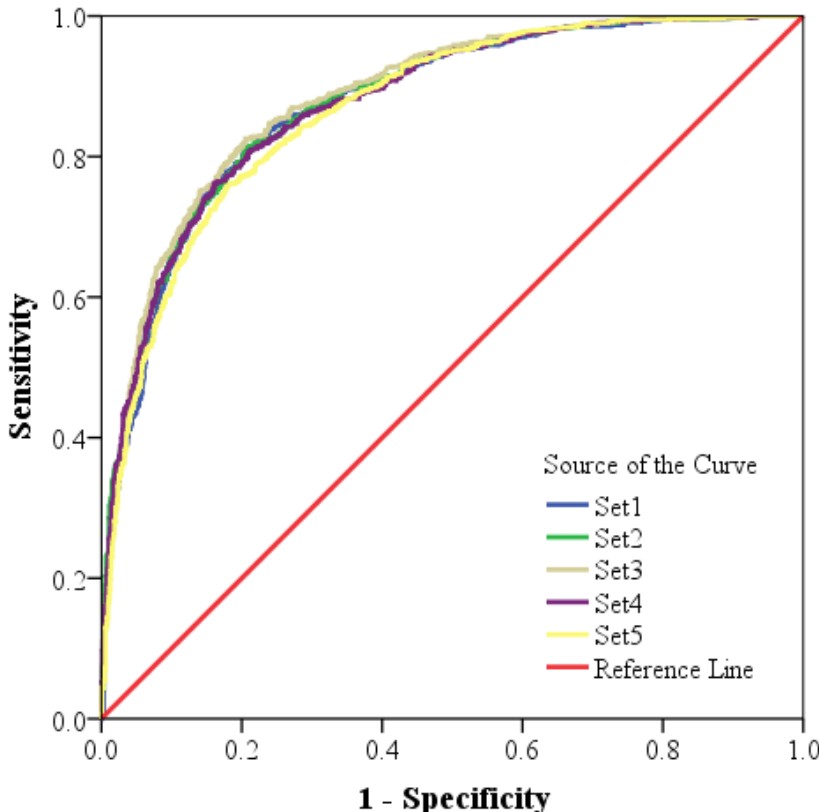

3

1    **Fig.4** ROC curve of validation process

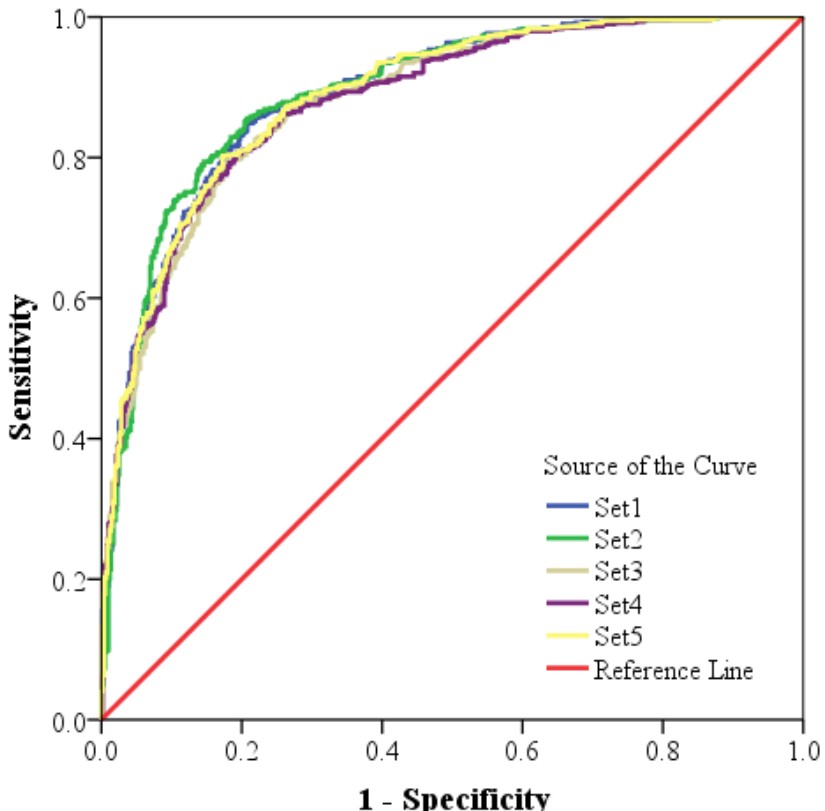

4

1      **Fig.5** Global-scale landslide susceptibility map

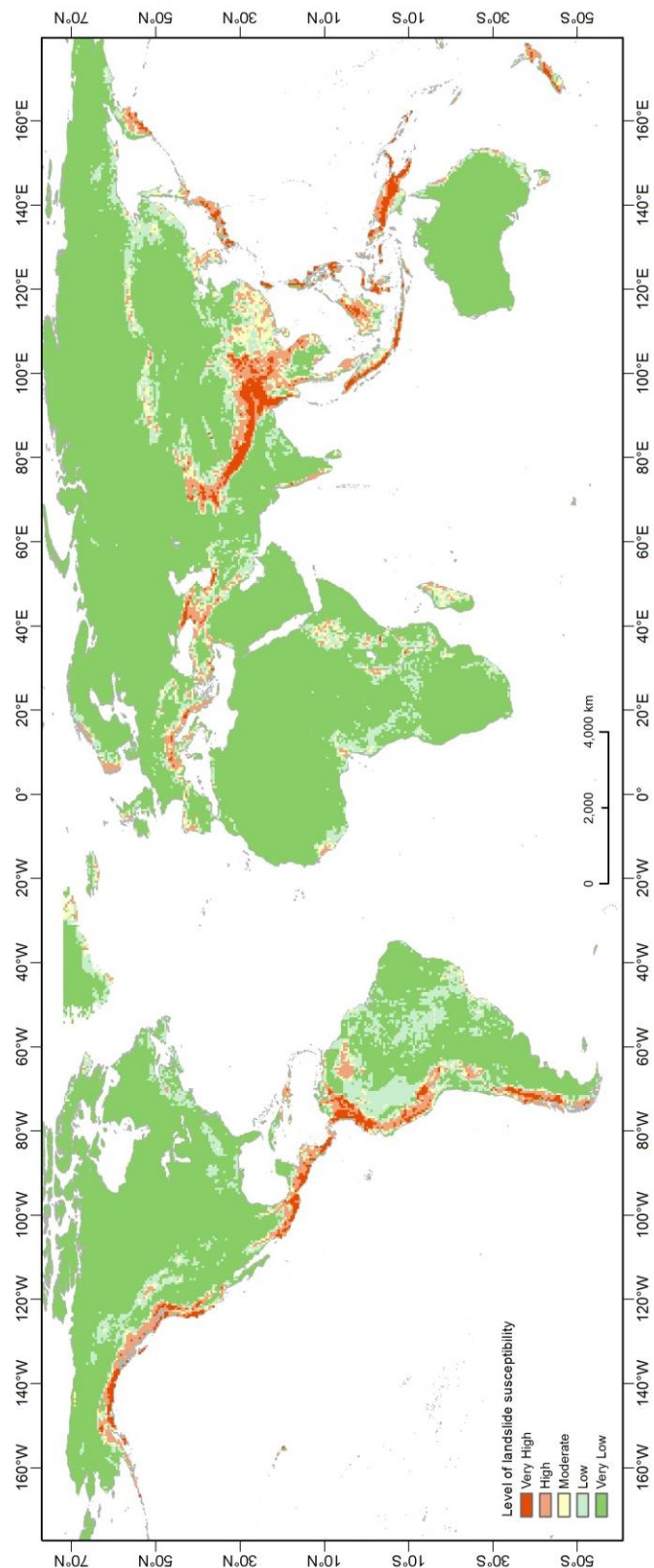

     **Fig. 6** Comparison of existing studies with the related parts of this study

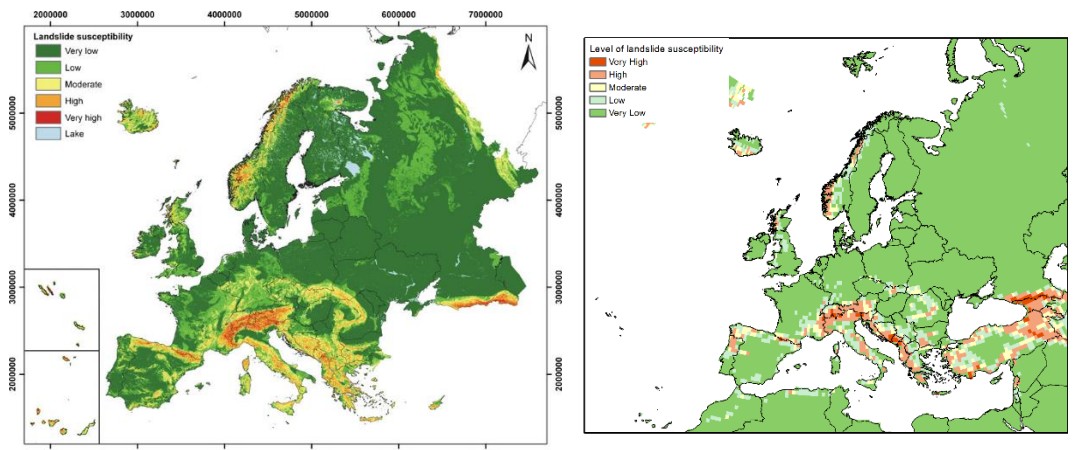

(a) Comparison of European landslide susceptibility map (from Van Den Eeckhaut et al. 2012) with
the related part in this study's map

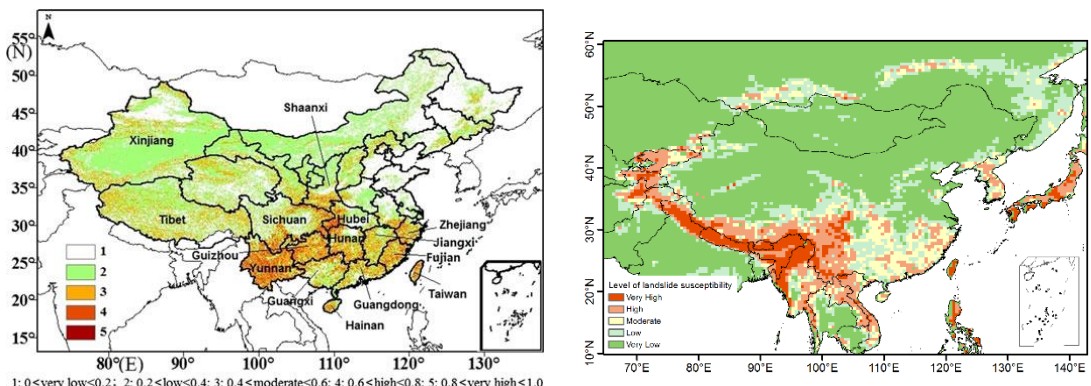

(b) Comparison of China landslide susceptibility map (from Liu et al. 2013)
with the related part in this study's map

