# Peer review of "Landslide susceptibility mapping on global scale using method"

_Natural Hazards and Earth System Sciences, 2016_

## Referee Comment (RC1) · Anonymous Referee #1 · 29 Dec 2016

General Comments

The manuscript "Landslide susceptibility mapping on global scale using method of logistic regression" aims to produce a global landslide susceptibility map, thematic that fits the scope of NHESS journal. Methodological framework is the major strength of the presented work with clear description of the used "explanatory factors" and validation of the obtained results. However, it is not the case for the dependent variable (landslide inventory) which need deep clarifications of data base "procedures" and tests to their representativeness. Unbalanced manuscript when comparing Methodology and data section with Results and Discussion sections. Methodology should be increased with inventories information and Results and Discussion sections, with a clear highlight of major assumptions and uncertainties, should be much more developed.

Specific comments

1) General state of the art is well developed. However some references to works that deal with large areas inventories in Europe should be done (e.g. Van Den Eeckhaut, M., Hervás, J., 2012. State of the art of national landslide databases in Europe and their potential for assessing susceptibility, hazard and risk. Geomorphology 139 (140), 545–558.

2) In reviewer opinion no novelty is presented considering concepts, ideas, data or methods. Despite the reference that global landslide maps can be used by industries, NGO or international cooperation, the authors should make clear what the added values of their study are. How a global landslide susceptibility map will be used for insurances that secure buildings in a local scale? Why the use of logistic regression which allow to weight each factor is importante if a world map is presented and those weights will not be used in other areas?

3)The "inventory" section is in reviewer opinion the major weakness of the presented manuscript Two general databases are combined and used: World Geological Hazard Inventory and NASA global landslide inventory. However, despite the references to the original sources some ideas should be clear in this manuscript as for example: a) what are the criteria that were used to recognize a landslide or to be inserted in the database; b) what is the best resolution?; c) are criteria/resolution the same in both inventories? d) Sometimes that kind of inventories have a high degree of uncertainty in location. How you deal with "overlapping" of registries? e) What area the time-period of those inventories? f) It is not clear for me if authors (or the team of 10 persons) collect information in newspapers? And the literature what are the main sources? only peer-review journals? Thesis? From which editors or universities? How many references are considered? Cross reference problems?

4) A deep discussion should be done about how representative are these inventories. According to the authors 2005 landslides/debris flows are in the database, but for ex-

ample in the work of Pereira et al. (2014) which use a historical landslide inventory based on press and reports in the northern region of a small country as Portugal, more than 600 cases were registered. Are the authors confident with 2005 registries for a World Wide Map? In my opinion this is a strong weakness of this work.

Pereira S, Zêzere JL, Quaresma ID, Bateira C (2014) Landslide incidence in the North of Portugal: Analysis of a historical landslide database based on press releases and technical reports. Geomorphology 214:514–525. doi: 10.1016/j.geomorph.2014.02.032

5) Authors should try to compare subsets of their inventory with other national or "continental" ones (with higher detail and available in literature) to try to find if spatial overlay is acceptable. Some metric should be done;

6) Considering figure 1 it seems that some areas are overestimated and that could be the result of the used inventories. In fact most of the dots are in Asia, and I believe that could be true, but it should be supported with statistical data from international databases, for example EM-DAT, used by UN. It seems to me that North Africa mountain ranges are underestimated;

7) A table with the number of landslides per region (for example continent or other wide regions that the authors consider adequate) in global inventory and in each data set used to model and validate will allow the reader to understand the how spatial representative are the data sets used. This should be inserted in results section;

8) The first paragraph of Results section is mainly methodological procedure for validation.

9) Maybe the used inventories are biased by the scale of analysis and the adopted recognition methodology (small movements disappear) or by the used criteria to consider a landslide (for example only landslides that cause injuries). This and other assumptions related to the inventories should be deeply discussed in Discussion section;

[Figure]

9) Even if only a visual evaluation is possible to do: how different are the obtained results when compared with other global landslide susceptibility maps (some of them referred in this work)? And with other national/regional maps (for example, USA or Europe (Join Research Center))?

Technical corrections

Page 3 line 33; Page 7 line 26 – please confirm the use of the term "topology". Topography?

Please check the way how you performed in-text citations to several references: sometimes they are alphabetically (e.g. page 2 line 5), others chronologically (e.g. page 2 line 21) and others none of them (page 2 lines 17-18; page 3 line 25; page 4lines 15-16; page 5 lines 39-40; page 6 lines 14-15). Figures are adequate but in general with low resolution.

Figure 1 (in figure or caption) should include time-period of the inventory and a reference to the main sources of the inventory. Each dot (triangle) means 1 landslide or more? Please provide a similar graphical scale in figures 1 and 2. I suggest 0-2000 km.

---

## Referee Comment (RC2) · Anonymous Referee #2 · 12 Feb 2017

General Comments

Global landslide mapping is important for disaster mitigation and prevention. This manuscript fits the scope of NHESS journal well. The structure is good and the logic flow can be easily followed. However, the discussion and other parts of this work should be improved.

Specific comments

1) By considering landslide triggering factors, this work is more like a qualitative hazard mapping rather than a susceptibility mapping (van Westen, et al., 2008; Nadim et al., 2008; Fell et al., 2008).

van Westen, C. J., et al. (2008). "Spatial data for landslide susceptibility, hazard, and

vulnerability assessment: An overview." Engineering Geology 102(3-4): 112-131.

Nadim, F., et al. (2006). "Global landslide and avalanche hotspots." Landslides 3(2): 159-173.

Fell, R., et al. (2008). "Guidelines for landslide susceptibility, hazard and risk zoning for land use planning." Engineering Geology 102(3–4): 85-98.

2) The slope gradient factor should be added, which is as important as relative relief. Because, it is a common sense that steeper slopes are easier to have landslides than gentler ones.

3) In addition, land cover is also an important influencing factor on landslide susceptibility mapping. It is well acknowledged that vegetation, especially trees can prevent some shallow landslides. The authors are suggested to consider land cover types in their mapping.

4) The authors used two datasets for dependent variable. Is there any consistency between them? Or, can you simply use them by combining both data sets? For example, maybe the Chinese datasets has more landslides within China while underestimate landslides abroad. Also, please introduce this new dataset in more detail, as there seems to be rare reports of it before.

5) An improved discussion is needed to compare and highlight the contribution of this work in global landslide mapping compared to previous works.

---

## Author Comment (AC1) · 14 Apr 2017

Response to Anonymous Referee #1

In this document, the underlined part is those revision we made for a new manuscript.

**Q1 (Question 1)**: General state of the art is well developed. However some references to works that deal with large areas inventories in Europe should be done (e.g. Van Den Eeckhaut, M., Hervás, J., 2012. State of the art of national landslide databases in Europe and their potential for assessing susceptibility, hazard and risk. Geomorphology 139 (140), 545–558.

**Response 1 (R1)**: Thank you for your comment. In the new manuscript, we have investigated more literatures and provided more about landslide databases in Europe and their potential for assessing susceptibility, hazard and risk. The following underlined part has been added in the new manuscript.

A landslide inventory provides the basis for quantitative zoning of landslide susceptibility. Location, date, type, size, causal factors and damage are supposed to be included in this database. A commonly used landslide inventory does not yet appear but some regional or national landslide databases are now well developed. In Europe, currently 22 out of 37 contacted countries have national landslide databases, and six other countries only have regional landslide databases. Those national databases contain about 633,700 landslides in total, of which about 75% are in Italy, and more than 10,000 landslides are in Austria, the Czech Republic, France, Norway, Poland, Slovakia, and the UK. In these 37 European countries, only six have sufficient information to perform risk analysis and one to perform a hazard analysis, while 14 countries can carry out at least a susceptibility analysis. Therefore, at a continental scale landslide zoning seems to be limited to landslide susceptibility modelling only. Restricted access to the data also make it difficult for these data to be applied in scientific research[1].

**Q2**: The authors should make clear what the added values of their study are. How a global landslide susceptibility map will be used for insurances that secure buildings in a local scale? Why the use of logistic regression which allow to weight each factor is important if a world map is presented and those weights will not be used in other areas?

**R2**: Thank you for your comment. The added value of this research is in the part of introduction and we have made it clearer. This paper addresses the gap in creating global landslide susceptibility maps using the widely used statistical method: logistic regression, and demonstrating the relative significance of different explanatory factors in global scale. The global landslide susceptibility map in this paper is not for securing local buildings actually. For those international and national insurance or reinsurance companies, such map will provide them with clear knowledge of landslide hotspots at a macro level, which will help them concentrate on those susceptible areas and make relevant marketing strategies like transferring risks. These underlined part has been added in the new manuscript.

Compared with some complex numerical methods like SVM, logistic regression provides a simple method to produce global landslide susceptibility map, which would

[1] Van Den Eeckhaut, M., & Hervás, J. (2012). State of the art of national landslide databases in Europe and their potential for assessing landslide susceptibility, hazard and risk. Geomorphology, 139, 545-558.

be helpful in disseminating this research and could encourage further model development for its simplicity in modelling. What's more, the result from logistic regression could illustrate the relative importance of different factors in explaining landslides, which could not be achieved by SVM. The accuracy of logistic regression model in this paper is quite high compared with that of similar experiment that is performed in national[2] or local[3] scale, which really exceeds expectation. To have one single model to explain the occurrence of past landslides events in global scale may be difficult, but the result of model in this paper shows that the factors and their weights in this research can actually provide good explanation of global landslide occurrence in one model.

This is the reason why the method of logistic regression is used and it has been added in the new manuscript.

As for the weights of factor, we have to agree that they cannot provide adequate accuracy when building landslide model in local scale. However, we have determined to compare this research with those performed in local scale to investigate the rules of landslide occurrence in different scales in the coming future. We have added this part in the discussions.

**Q3**: The "inventory" section is in reviewer opinion the major weakness of the presented manuscript. Two general databases are combined and used: World Geological Hazard Inventory and NASA global landslide inventory. However, despite the references to the original sources some ideas should be clear in this manuscript as for example: a) what are the criteria that were used to recognize a landslide or to be inserted in the database; b) what is the best resolution? c) are criteria/resolution the same in both inventories? d) Sometimes that kind of inventories have a high degree of uncertainty in location. How you deal with "overlapping" of registries? e) What area the time-period of those inventories? f) It is not clear for me if authors (or the team of 10 persons) collect information in newspapers? And the literature what are the main sources? only peer review journals? Thesis? From which editors or universities? How many references are considered? Cross reference problems?

**R3**: Thank you for your comment. The landslide data in this research comes from BNU World Geological Hazard Inventory and NASA global landslide inventory. In the former manuscript, we did not provide adequate information about it. We have added more in this manuscript:

(1)The sources of related databases. The entries of World Geological Hazard Inventory mainly come from news reports (e.g. mass media in China, Xinhua News, and Sina News) and records in books and journals. We searched information about landslide on Internet by using keywords like landslide and debris flow. Then we read these descriptions carefully to determine whether it is a landslide and locate it, and later put it into the database. Thus the main source of World Geological Hazard Inventory can be news data. By investigating these news, we can find out those

[2] Lin, Q., Wang, Y., Liu, T., Zhu, Y., & Sui, Q. (2017). The vulnerability of people to landslides: a case study on the relationship between the casualties and volume of landslides in China. International journal of environmental research and public health, 14(2), 212.

[3] Wang, Y., Song, C., Lin, Q., & Li, J. (2016). Occurrence probability assessment of earthquake-triggered landslides with Newmark displacement values and logistic regression: The Wenchuan earthquake, China. Geomorphology, 258, 108-119.

landslides that are of large volume or of high danger, for these kinds of landslides can be of high news value.

The NASA global landslide inventory mainly collects landslides from several existing databases, including International Consortium on Landslides website (ICL); International Landslide Centre, University of Durham (ILC); The EM-DAT International Disaster Database; International Federation of Red Cross and Red Crescent Societies field reports; Reliefweb; humanitarian disaster information run by the United Nations Office for the Coordination of Humanitarian Affairs (OCHA); other online regional and national newspaper articles and media sources.

The best resolution of World Geological Hazard Inventory is 0.001 degree, and the NASA global landslide inventory 2km.

(2) The time period of landslide database. In the World Geological Hazard Inventory, the earliest event can be dated to 1618. In this database, there is 117 landslides occurred before 1975, 84 between 1975 to 2000, and 274 between 2000 and 2014. The landslide events in the NASA global landslide inventory mainly happened in 2003, 2007, 2008 and 2009. Hence these two databases are complementary and they can be emerged to produce a more complete landslide database. This part has been added in the new manuscript.

(3) The combination of two databases. When combining these two databases, the occurrence of time provides crucial standard. When two landslide events have different time (month), they are both reserved in the new database. If two events have the same occurrence time (month) and their locations are close, investigation through details in source could determine whether they are from the same disaster. If yes, the record with higher spatial resolution is reserved and the one with lower resolution is dropped. This part has been added in the new manuscript.

**Q4**: A deep discussion should be done about how representative are these inventories. According to the authors 2005 landslides/debris flows are in the database, but for example in the work of Pereira et al. (2014) which use a historical landslide inventory based on press and reports in the northern region of a small country as Portugal, more than 600 cases were registered. Are the authors confident with 2005 registries for a World Wide Map? In my opinion this is a strong weakness of this work. Pereira S, Zêzere JL, Quaresma ID, Bateira C (2014) Landslide incidence in the North of Portugal: Analysis of a historical landslide database based on press releases and technical reports. Geomorphology 214:514–525. doi:10.1016/j.geomorph.2014.02.032

**R4**: Thank you for your comment. We have read the paper of Pereira et al. The landslide data in that paper comes from press releases (regional and local newspapers) and technical reports (reports by civil protection authorities and academic works). The database contains 628 landslides with time period of 1900 to 2010. Pereira et al. also mentioned that regional and local newspapers are more effective than the national newspaper in reporting damaging landslides in the North of Portugal.

Compared with the research of Pereira et al., the amount of landslide records in our research seems not enough. However, landslides may have different magnitudes. In the analysis of global landslide susceptibility, landslides in database should be representative and those landslides with large magnitudes can meet such requirement. The landslides in our database are either of large magnitudes or causing severe life

loss or economic loss. If we don't following this guidelines, in our database there will be a large numbers of landslides that are occurred in countries with good landslide catalogue and few in countries with poor landslide catalogue. Such model may lead to overestimation of landslide susceptibility in countries with rich landslide records and underestimation of landslide susceptibility in countries with poor landslide records. This may not be good for improving the accuracy of the map of global landslide susceptibility. Hence we think that the landslide database in our research is high representative and reliable. Thank you for your recommendation. We will explore the use of big data on Internet in building more comprehensive landslide database in our next research.

**Q5**: Authors should try to compare subsets of their inventory with other national or "continental" ones (with higher detail and available in literature) to try to find if spatial overlay is acceptable. Some metric should be done.

**R5**: Thank you for your comment. In our research, we have tried to acquire landslide databases of other nations or areas but failed. Hence we compare the spatial overlay of our landslide database with that of Europe[4].

[Figure]

From the above comparison, it is found that the spatial overlay of landslide samples in the research of European landslide susceptibility modelling is quite similar with that of the combined landslide database in this research. The landslides in Europe mainly distribute in mountainous areas like the Alps and the Balkan. These have been added in the new manuscript.

**Q6**: Considering figure 1 it seems that some areas are overestimated and that could be the result of the used inventories. In fact most of the dots are in Asia, and I believe that could be true, but it should be supported with statistical data from international databases, for example EM-DAT, used by UN. It seems to me that North Africa mountain ranges are underestimated;

A table with the number of landslides per region (for example continent or other wide regions that the authors consider adequate) in global inventory and in each data set
* * *
[4] Van Den Eeckhaut, M., Hervás, J., Jaedicke, C., Malet, J. P., Montanarella, L., & Nadim, F. (2012). Statistical modelling of Europe-wide landslide susceptibility using limited landslide inventory data. Landslides, 9(3), 357-369.

used to model and validate will allow the reader to understand the how spatial representative are the data sets used. This should be inserted in results section;

**R6**: Thank you for your comment. In fact, the landslides in EM-DAT has been contained in our database. It is because EM-DAT is one important source of the NASA global landslide inventory and also the World Geological Hazard Inventory.

As for the samples in North Africa, thank you for reminding us on such issue. We are now checking our data again and recalculate this. Unfortunately, we cannot finish this part by now. We promise to upload our modelling result of each continent in next week on the open discussion of the journal of NHESS.

**Q7**: The first paragraph of Results section is mainly methodological procedure for validation.

**R7**: Thank you for your comment. We agree that this part should be replaced and we have adjusted it in the new manuscript.

**Q8**: Maybe the used inventories are biased by the scale of analysis and the adopted recognition methodology (small movements disappear) or by the used criteria to consider a landslide (for example only landslides that cause injuries). This and other assumptions related to the inventories should be deeply discussed in Discussion section;

**R8**: Thank you for your comment. We have added these following underlined part in discussions. Because of the main focus of this research is global landslide susceptibility assessment, the landslides in database of this research are mainly those of large volume or causing significant loss, which are hence easily reported by news agencies. In such background, some landslides with small magnitudes or not causing loss may be ignored. The global landslide susceptibility map built on this database can inevitably underestimate the landslide susceptibility in some sparsely populated areas or less developed areas. At the same time, besides the factors studied in this paper, there can be some other factors that also influence the occurrence of landslides. This research intends to propose a more quantitative method in assessing landslide susceptibility and tries to enhance the studies of landslide susceptibility when landslide catalogues from various countries can be easily accessed in the future.

**Q9**: Even if only a visual evaluation is possible to do: how different are the obtained results when compared with other global landslide susceptibility maps (some of them referred in this work)? And with other national/regional maps (for example, USA or Europe (Join Research Center))?

**R9**: Thank you for your comment. We have added relevant comparison as follows.

The global landslide susceptibility map may be evaluated by comparison with four studies from the current literature that focus on large-scale landslide susceptibility.

Comparing the European landslide susceptibility map drawn by Van Den Eeckhaut et al. (2012)[5] with the European part of susceptibility map in this study (Fig. 5 (a)), similar areas of high landslide susceptibility can be observed. The former map includes two levels (denoted High and Very High) as high susceptibility with a landslide probability of over 0.8, and this study also includes two levels (Levels 4 and 5) as high susceptibility with a probability over 0.7. The two maps have similar high susceptibility areas. Thus, for Europe, landslide susceptibility map in this study agrees with existing related study.

[Figure]

Fig. 5 (a) Comparison of European landslide susceptibility map (from Van Den Eeckhaut et al. 2012) with the related part in this study's map

[Figure]

Fig. 5 (b) Comparison of China landslide susceptibility map (from Liu et al. 2013) with the related part in this study's map

Comparing the Chinese landslide susceptibility map drawn by Liu et al. (2013)[6] with the China part of susceptibility map in this study (Fig. 5 (b)), the former map includes two levels (Levels 4 and 5) as susceptible with a landslide probability of over 0.6. Map in this study includes three levels (denoted Levels 3, 4 and 5) as susceptible with landslide probability of over 0.6. The main differences between the two maps are in the western Sichuan Basin and southern Tibet, which is famous for its high elevation and intense relative relief. This study applies many landslide cases in these areas.

[5] Van Den Eeckhaut, M., Hervás, J., Jaedicke, C., Malet, J. P., Montanarella, L., & Nadim, F. (2012). Statistical modelling of Europe-wide landslide susceptibility using limited landslide inventory data. Landslides, 9(3), 357-369.

[6] Liu, C., Li, W., Wu, H., Lu, P., Sang, K., Sun, W., & Li, R. (2013). Susceptibility evaluation and mapping of China's landslides based on multi-source data. Natural hazards, 69(3), 1477-1495.

However, in the landslide database of Liu et al. (2013), only a few landslides occur in these areas. This discrepancy is the reason for the differences between the two maps.

As for landslide susceptibility at global scale, Nadim et al. (2006) and Hong et al. (2007) have ever made magnificent efforts on such topic. One global landslide susceptibility map (please refer to Fig. 7 in Nadim et al. (2006)) has five levels (Levels 5, 6, 7, 8 and 9) as susceptible, while the map from this study includes three levels (Levels 3, 4 and 5) as susceptible. In general, the susceptible areas of these two maps are fairly similar except in Madagascar and the eastern Indo-China Peninsula.

Another global landslide susceptibility map (please refer to Fig. 3(a) in Hong et al. (2007)) has two levels (Levels 4 and 5) as susceptible, compared to map in this study, which has three levels (Levels 3, 4 and 5) as susceptible. These two maps are similar over Asia, Europe and Africa. However, map from Hong et al. (2007) has more details over the Americas, for example, showing landslide susceptible areas in the Appalachian Mountains in North America and in the Brazilian Highlands in South America. To a large extent, this greater detail occurs because Hong et al. (2007) used landslide susceptibility map of the United States to adjust the combination weights of explanatory factors in their global model. It is noted that map of Hong et al. (2007) also differs from map of this study in that it shows high landslide susceptibility in central and southern India, and low landslide susceptibility in equatorial islands such as Malaysia, Indonesia, and the Philippines.

**Q10**: Page 3 line 33; Page 7 line 26 – please confirm the use of the term "topology". Topography? Please check the way how you performed in-text citations to several references: sometimes they are alphabetically (e.g. page 2 line 5), others chronologically (e.g. page 2 line 21) and others none of them (page 2 lines 17-18; page 3 line 25; page 4 lines 15-16; page 5 lines 39-40; page 6 lines 14-15). Figures are adequate but in general with low resolution.

**R10**: Thank you for your comment. We have revised those parts mentioned above. In the new manuscript, we will provide figures with higher resolution. You can also find them in the attachment of this response.

**Q11**: Figure 1 (in figure or caption) should include time-period of the inventory and a reference to the main sources of the inventory. Each dot (triangle) means 1 landslide or more? Please provide a similar graphical scale in figures 1 and 2. I suggest 0-2000 km.

**R11**: Thank you for your comment. The time period and data reference have been added in Figure 1. Each dot represents one landslide. Similar graphical scale in Figures 1 and 2 has also been added. Please check the Figure 1 & 2 document in the attachment of this response.

---

## Author Comment (AC2) · 14 Apr 2017

Response to Anonymous Referee #2

In this document, the underlined part is those revision we made for a new manuscript.

**Question 1 (Q1)**: By considering landslide triggering factors, this work is more like a qualitative hazard mapping rather than a susceptibility mapping (van Westen, et al., 2008; Nadim et al., 2008; Fell et al., 2008). van Westen, C. J., et al. (2008). "Spatial data for landslide susceptibility, hazard, and vulnerability assessment: An overview." Engineering Geology 102(3-4): 112-131. Nadim, F., et al. (2006). "Global landslide and avalanche hotspots." Landslides 3(2): 159-173. Fell, R., et al. (2008). "Guidelines for landslide susceptibility, hazard and risk zoning for land use planning." Engineering Geology 102(3–4): 85-98.

**Response 1 (R1)**: Thank you for your comment. In the paper of van Westen et al. (2008), hazard assessment should include temporal and spatial probability of initiation, magnitude–frequency relation and run out potential. Because we did not study the temporal aspect of landslides, we hence did not put qualitative hazard mapping in the title. However, as we put some triggering factors in models, we tend to agree with your comment. The title of this study is corrected to Landslide hazard mapping on global scale using method of logistic regression and related revisions have been made in the new manuscript.

**Q2**: The slope gradient factor should be added, which is as important as relative relief. Because, it is a common sense that steeper slopes are easier to have landslides than gentler ones.

**R2**: Thank you for your comment. We agree that slope is very important factor in the research of landslide susceptibility. We have included this factor when building landslide model. But the result show that it is not statistically significant. Therefore we did not include it in this paper. We have analysed the reason. At a global scale, factors such as elevation and slope gradient can be replaced by topographic index or relative relief, which indicate macroscopic differences in topography. Especially for landslide data with low location precision, using factors such as elevation or slope gradient that precisely relate to landslide location will reduce the accuracy of landslide susceptibility analysis. This part has been added in the new manuscript.

**Q3**: In addition, land cover is also an important influencing factor on landslide susceptibility mapping. It is well acknowledged that vegetation, especially trees can prevent some shallow landslides. The authors are suggested to consider land cover types in their mapping.

**R3**: Thank you for your comment. Like the factor of slope, we included the factor of land cover when performing experiments. The land cover product with spatial resolution of 30m, GlobeLand30[1], is produced by scientists in China and submitted to United Nations for public use[2]. We tried this factor but found that it is not statistically significant and does not improve the model accuracy. Hence the factor of land cover is not included in this paper. The reason can be that comparing with other factors, land cover may not be a significantly important factor in assessing landslide susceptibility in global scale.

**Q4**: The authors used two datasets for dependent variable. Is there any consistency between them? Or, can you simply use them by combining both data sets? For example, maybe the Chinese datasets has more landslides within China while underestimate landslides abroad. Also, please introduce this new dataset in more detail, as there seems to be rare reports of it before.

**R4**: Thank you for your comment. The landslide data in this research comes from BNU World Geological Hazard Inventory and NASA global landslide inventory. In the former manuscript, we did not provide adequate information about it. We have added more in this manuscript:

(1)The sources of related databases. The entries of World Geological Hazard Inventory mainly come from news reports (e.g. mass media in China, Xinhua News, and Sina News) and records in books and journals. We searched information about landslide on Internet by using keywords like landslide and debris flow. Then we read these descriptions carefully to determine whether it is a landslide and locate it, and later put it into the database. Thus the main source of World Geological Hazard Inventory can be news data. By investigating these news, we can find out those landslides that are of large volume or of high danger, for these kinds of landslides can be of high news value.

The NASA global landslide inventory mainly collects landslides from several existing databases, including International Consortium on Landslides website (ICL); International Landslide Centre, University of Durham (ILC); The EM-DAT International Disaster Database; International Federation of Red Cross and Red Crescent Societies field reports; Reliefweb; humanitarian disaster information run by the United Nations Office for the Coordination of Humanitarian Affairs (OCHA); other online regional and national newspaper articles and media sources.
* * *
[1] http://www.globeland30.org/GLC30Download/index.aspx

[2] https://unstats.un.org/unsd/GlobeLand30.htm

The best resolution of World Geological Hazard Inventory is 0.001 degree, and the NASA global landslide inventory 2km.

(2) The time period of landslide database. In the World Geological Hazard Inventory, the earliest event can be dated to 1618. In this database, there is 117 landslides occurred before 1975, 84 between 1975 to 2000, and 274 between 2000 and 2014. The landslide events in the NASA global landslide inventory mainly happened in 2003, 2007, 2008 and 2009. Hence these two databases are complementary and they can be emerged to produce a more complete landslide database. This part has been added in the new manuscript.

(3) The combination of two databases. When combining these two databases, the occurrence of time provides crucial standard. When two landslide events have different time (month), they are both reserved in the new database. If two events have the same occurrence time (month) and their locations are close, investigation through details in source could determine whether they are from the same disaster. If yes, the record with higher spatial resolution is reserved and the one with lower resolution is dropped. This part has been added in the new manuscript.

**Q5**: An improved discussion is needed to compare and highlight the contribution of this work in global landslide mapping compared to previous works.

**R5**: Thank you for your comment. We have added relevant comparison as follows.

The global landslide susceptibility map may be evaluated by comparison with four studies from the current literature that focus on large-scale landslide susceptibility.

Comparing the European landslide susceptibility map drawn by Van Den Eeckhaut et al. (2012)[3] with the European part of susceptibility map in this study (Fig. 5 (a)), similar areas of high landslide susceptibility can be observed. The former map includes two levels (denoted High and Very High) as high susceptibility with a landslide probability of over 0.8, and this study also includes two levels (Levels 4 and 5) as high susceptibility with a probability over 0.7. The two maps have similar high susceptibility areas. Thus, for Europe, landslide susceptibility map in this study agrees with existing related study.
* * *
[3] Van Den Eeckhaut, M., Hervás, J., Jaedicke, C., Malet, J. P., Montanarella, L., & Nadim, F. (2012). Statistical modelling of Europe-wide landslide susceptibility using limited landslide inventory data. Landslides, 9(3), 357-369.

[Figure]

Fig. 5 (a) Comparison of European landslide susceptibility map (from Van Den Eeckhaut et al. 2012) with the related part in this study's map

[Figure]

Fig. 5 (b) Comparison of China landslide susceptibility map (from Liu et al. 2013) with the related part in this study's map

Comparing the Chinese landslide susceptibility map drawn by Liu et al. (2013)[4] with the China part of susceptibility map in this study (Fig. 5 (b)), the former map includes two levels (Levels 4 and 5) as susceptible with a landslide probability of over 0.6. Map in this study includes three levels (denoted Levels 3, 4 and 5) as susceptible with landslide probability of over 0.6. The main differences between the two maps are in the western Sichuan Basin and southern Tibet, which is famous for its high elevation and intense relative relief. This study applies many landslide cases in these areas. However, in the landslide database of Liu et al. (2013), only a few landslides occur in these areas. This discrepancy is the reason for the differences between the two maps.

As for landslide susceptibility at global scale, Nadim et al. (2006) and Hong et al. (2007) have ever made magnificent efforts on such topic. One global landslide susceptibility map (please refer to Fig. 7 in Nadim et al. (2006)) has five levels (Levels 5, 6, 7, 8 and 9) as susceptible, while the map from this study includes three levels (Levels 3, 4 and
* * *
[4] Liu, C., Li, W., Wu, H., Lu, P., Sang, K., Sun, W., & Li, R. (2013). Susceptibility evaluation and mapping of China's landslides based on multi-source data. Natural hazards, 69(3), 1477-1495.

5) as susceptible. In general, the susceptible areas of these two maps are fairly similar except in Madagascar and the eastern Indo-China Peninsula.

Another global landslide susceptibility map (please refer to Fig. 3(a) in Hong et al. (2007)) has two levels (Levels 4 and 5) as susceptible, compared to map in this study, which has three levels (Levels 3, 4 and 5) as susceptible. These two maps are similar over Asia, Europe and Africa. However, map from Hong et al. (2007) has more details over the Americas, for example, showing landslide susceptible areas in the Appalachian Mountains in North America and in the Brazilian Highlands in South America. To a large extent, this greater detail occurs because Hong et al. (2007) used landslide susceptibility map of the United States to adjust the combination weights of explanatory factors in their global model. It is noted that map of Hong et al. (2007) also differs from map of this study in that it shows high landslide susceptibility in central and southern India, and low landslide susceptibility in equatorial islands such as Malaysia, Indonesia, and the Philippines.

---

## Author Response (AR1)

**Response to Anonymous Referee #1**
In this document, the underlined part is those revision we made for a new manuscript.

**Q1 (Question 1)**: General state of the art is well developed. However some references to works that deal with large areas inventories in Europe should be done (e.g. Van Den Eeckhaut, M., Hervás, J., 2012. State of the art of national landslide databases in Europe and their potential for assessing susceptibility, hazard and risk. Geomorphology 139 (140), 545–558.

**Response 1 (R1)**: Thank you for your comment. In the new manuscript, we have investigated more literatures and provided more about landslide databases in Europe and their potential for assessing susceptibility, hazard and risk. The following underlined part has been added in the new manuscript.

A landslide inventory provides the basis for quantitative zoning of landslide susceptibility. Location, date, type, size, causal factors and damage are supposed to be included in this database. A commonly used landslide inventory does not yet appear but some regional or national landslide databases are now well developed. In Europe, currently 22 out of 37 contacted countries have national landslide databases, and six other countries only have regional landslide databases. Those national databases contain about 633,700 landslides in total, of which about 75% are in Italy, and more than 10,000 landslides are in Austria, the Czech Republic, France, Norway, Poland, Slovakia, and the UK. In these 37 European countries, only six have sufficient information to perform risk analysis and one to perform a hazard analysis, while 14 countries can carry out at least a susceptibility analysis. Therefore, at a continental scale landslide zoning seems to be limited to landslide susceptibility modelling only. Restricted access to the data also make it difficult for these data to be applied in scientific research. (Pg 2 Ln 25-36, the number in the clear version uploaded)

**Q2**: The authors should make clear what the added values of their study are. How a global landslide susceptibility map will be used for insurances that secure buildings in a local scale? Why the use of logistic regression which allow to weight each factor is important if a world map is presented and those weights will not be used in other areas?

**R2**: Thank you for your comment. The added value of this research is in the part of introduction and we have made it clearer. This paper addresses the gap in creating global landslide susceptibility maps using the widely used statistical method: logistic regression, and demonstrating the relative significance of different explanatory factors in global scale.(Pg 3 Ln 13-15) The global landslide susceptibility map in this paper is not for securing local buildings actually. For those international and national insurance or reinsurance companies, such map will provide them with clear knowledge of landslide hotspots at a macro level, which will help them concentrate on those susceptible areas and make relevant marketing strategies like transferring risks. (Pg 1 Ln 35-38) These underlined part has been added in the new manuscript.

Compared with some complex numerical methods like SVM, logistic regression provides a simple method to produce global landslide susceptibility map, which would be helpful in disseminating this research and could encourage further model development for its simplicity in modelling. What's more, the result from logistic regression could illustrate the relative importance of different factors in explaining landslides, which could not be achieved by SVM. (Pg 3 Ln 7-12) The accuracy of logistic regression model in this paper is quite high compared with that of similar experiment that is performed at national or local scale, which really exceeds expectation. To have one single model to explain the occurrence of past landslides events in global scale may be difficult, but the result of model in this paper shows that the factors and their weights in this research can actually provide good explanation of global landslide occurrence in one model.(Pg 9 Ln 21-26)

This is the reason why the method of logistic regression is used and it has been added in the new manuscript.

As for the weights of factor, actually they cannot provide adequate accuracy when building landslide model in local scale. However, we have determined to compare this research with those performed in local scale to investigate the rules of landslide occurrence in different scales in the coming future. (Pg 9 Ln 30-33) We have added this part in the discussions.

**Q3**: The "inventory" section is in reviewer opinion the major weakness of the presented manuscript. Two general databases are combined and used: World Geological Hazard Inventory and NASA global landslide inventory. However, despite the references to the original sources some ideas should be clear in this manuscript as for example: a) what are the criteria that were used to recognize a landslide or to be inserted in the database; b) what is the best resolution? c) are criteria/resolution the same in both inventories? d) Sometimes that kind of inventories have a high degree of uncertainty in location. How you deal with "overlapping" of registries? e) What area the time-period of those inventories? f) It is not clear for me if authors (or the team of 10 persons) collect information in newspapers? And the literature what are the main sources? only peer review journals? Thesis? From which editors or universities? How many references are considered? Cross reference problems?

**R3**: Thank you for your comment. The landslide data in this research comes from BNU World Geological Hazard Inventory and NASA global landslide inventory. In the former manuscript, we did not provide adequate information about it. We have added more in this manuscript:

(1)The sources of related databases. The items in World Geological Hazard Inventory were collected manually from news reports (e.g. mass media in China, Xinhua News, and Sina News) and records in books and journals. We searched information about landslide on Internet by using keywords like landslide and debris flow. Then we read these descriptions carefully to determine whether it is a landslide and locate it, and later put it into the database. Thus the main source of World Geological Hazard Inventory can be news data. By investigating these news, we can find out those landslides that are of large volume or of high danger, for these kinds of landslides can be of high news value. (Pg 6 Ln 16-23)
The NASA global landslide inventory mainly collects landslides from several existing databases, including International Consortium on Landslides website (ICL; http://iclhq.org); International Landslide Centre, University of Durham (ILC; http://www.landslidecentre.org); The EM-DAT International Disaster Database (http://www.em-dat.net); International Federation of Red Cross and Red Crescent Societies field reports (http://www.ifrc.org); Reliefweb (http://reliefweb.int); humanitarian disaster information run by the United Nations Office for the Coordination of Humanitarian Affairs (OCHA); other online regional and national newspaper articles and media sources.(Pg 6 Ln 8-15)
The best resolution of World Geological Hazard Inventory is 0.001 degree, about 100m. (Pg 6 Ln 27) The NASA global landslide inventory 2km. (Pg 6 Ln 15-16)
(2) The time period of landslide database. In the World Geological Hazard Inventory, the earliest event can be dated to 1618. In this database, there is 117 landslides occurred before 1975, 84 between 1975 to 2000, and 274 between 2000 and 2014. The landslide events in the NASA global landslide inventory mainly happened in 2003, 2007, 2008 and 2009. Hence these two databases are complementary and they can be emerged to produce a more complete landslide database.(Pg 6 Ln 36-40) This part has been added in the new manuscript.
(3) The combination of two databases. When combining these two databases, the occurrence of time provides crucial standard. When two landslide events have different time (month), they are both reserved in the new database. If two events have the same occurrence time (month) and their locations are close, investigation through details in source could determine whether they are from the same disaster. If yes, the record with higher spatial resolution is reserved and the one with lower resolution is dropped. (Pg 6 Ln 30-34) This part has been added in the new manuscript.

**Q4**: A deep discussion should be done about how representative are these inventories. According to the authors 2005 landslides/debris flows are in the database, but for example in the work of Pereira et al. (2014) which use a historical landslide inventory based on press and reports in the northern region of a small country as Portugal, more than 600 cases were registered. Are the authors confident with 2005 registries for a World Wide

Map? In my opinion this is a strong weakness of this work. Pereira S, Zêzere JL, Quaresma ID, Bateira C (2014) Landslide incidence in the North of Portugal: Analysis of a historical landslide database based on press releases and technical reports. Geomorphology 214:514–525. doi:10.1016/j.geomorph.2014.02.032

**R4**: Thank you for your comment. We have read the paper of Pereira et al. The landslide data in that paper comes from press releases (regional and local newspapers) and technical reports (reports by civil protection authorities and academic works). The database contains 628 landslides with time period of 1900 to 2010. Pereira et al. also mentioned that regional and local newspapers are more effective than the national newspaper in reporting damaging landslides in the North of Portugal.

Compared with the research of Pereira et al., the amount of landslide records in our research seems not enough. However, landslides may have different magnitudes. The main focus of this research is global landslide susceptibility assessment, and hence the landslides in database of this research should be representative on global scale, i.e. having large volume or causing significant loss. The landslides in our database can meet such requirement and are either of large magnitudes or causing severe life loss or economic loss, which are hence easily reported by news agencies. The global landslide susceptibility map built on this database can inevitably underestimate the landslide susceptibility in some sparsely populated areas or less developed areas. However, if we don't following the guidelines, in our database there will be a large numbers of landslides that are occurred in countries with good landslide catalogue and few in countries with poor landslide catalogue. Such model may lead to overestimation of landslide susceptibility in countries with rich landslide records and underestimation of landslide susceptibility in countries with poor landslide records. This may not be good for improving the accuracy of the map of global landslide susceptibility. Hence we think that the landslide database in our research is relatively high in representative and reliable. We will explore the use of big data on Internet in building more comprehensive landslide database in our next research and tries to enhance the studies of landslide susceptibility when landslide catalogues from various countries can be easily accessed in the future. (Pg 9 Ln 34- Pg 10 Ln 4) Thank you for your recommendation.

**Q5**: Authors should try to compare subsets of their inventory with other national or "continental" ones (with higher detail and available in literature) to try to find if spatial overlay is acceptable. Some metric should be done.

**R5**: Thank you for your comment. These have been added in the new manuscript. In order to demonstrate the representative of landslide data used in this research, the landslide overlay in Europe of this research is compared with the spatial distribution of landslides in the study of Van Den Eeckhaut et al. (2012). As showed in Fig. 2, it is found that the spatial overlay of landslide samples in the research of European landslide susceptibility modelling is quite similar with that of the combined landslide database in this research. The landslides in Europe mainly distribute in mountainous areas like the Alps and the Balkan. (Pg 7 Ln 1-6)

**Fig.2** Comparison of landslide overlay in this research and that in existing study

[Figure]

[Figure]

**Q6**: Considering figure 1 it seems that some areas are overestimated and that could be the result of the used inventories. In fact most of the dots are in Asia, and I believe that could be true, but it should be supported with statistical data from international databases, for example EM-DAT, used by UN. It seems to me that North Africa mountain ranges are underestimated;

A table with the number of landslides per region (for example continent or other wide regions that the authors consider adequate) in global inventory and in each data set used to model and validate will allow the reader to understand the how spatial representative are the data sets used. This should be inserted in results section;

**R6**: Thank you for your comment. In fact, the landslides in EM-DAT has been contained in our database. It is because EM-DAT is one important source of the NASA global landslide inventory and also the World Geological Hazard Inventory.

As for the samples in North Africa, thank you for reminding us on such issue. We have added a table in the part of results to illustrate the spatial representative of landslides in each continent in each dataset.

A table with the number of landslides in each continent in global inventory and in each data set used to model and validate is displayed, which will help readers understand how spatial representative the data sets used are (Table 5).It can be found that there is a small amount of landslide records in Africa. However, when either in the modelling process or validation process, different amount of landslides and non-landslides in African was selected. From Fig. 3 and Fig. 4, it is demonstrated that the results from every five datasets are relatively stable and high, which means the model built can be applied effectively in Africa. Otherwise, the results of five datasets may be different. (Pg 8 Ln 7-13)

**Table 5** Numbers of landslides and non-landslides in each dataset.

| Continent | Landslides | Dataset 1 | | Dataset 2 | | Dataset 3 | | Dataset 4 | | Dataset 5 | |
|---|---|---|---|---|---|---|---|---|---|---|---|
| | | Modelling (70%) | Validation (30%) | Modelling (70%) | Validation (30%) | Modelling (70%) | Validation (30%) | Modelling (70%) | Validation (30%) | Modelling (70%) | Validation (30%) |
| Asia | 1205 | 847:348 | 358:163 | 848:394 | 357:157 | 838:383 | 367:155 | 849:364 | 356:162 | 847:393 | 358:165 |
| Africa | 69 | 55:317 | 14:129 | 50:307 | 19:114 | 47:324 | 22:130 | 47:315 | 22:129 | 49:302 | 20:126 |
| Europe | 121 | 94:211 | 27:70 | 86:212 | 35:116 | 87:200 | 34:91 | 88:206 | 33:88 | 85:251 | 36:106 |
| North America | 425 | 274:235 | 151:98 | 296:235 | 129:98 | 298:226 | 127:107 | 286:230 | 139:110 | 286:195 | 139:83 |
| South America | 133 | 93:189 | 40:98 | 86:174 | 47:75 | 99:179 | 34:78 | 97:193 | 36:80 | 96:144 | 37:64 |
| Oceania | 52 | 40:103 | 12:44 | 37:81 | 15:42 | 34:91 | 18:41 | 36:95 | 16:33 | 40:118 | 12:58 |
| Total | 2005 | 1403:1403 | 602:602 | 1403:1403 | 602:602 | 1403:1403 | 602:602 | 1403:1403 | 602:602 | 1403:1403 | 602:602 |

Numbers in left represent numbers of landslides, numbers in right represents numbers of non-landslides.

**Q7**: The first paragraph of Results section is mainly methodological procedure for validation.

**R7**: Thank you for your comment. We agree that this part should be replaced and we have adjusted it in the end of methodology and data in the new manuscript. Please refer to (Pg 7 Ln 22-27) in the newly revised manuscript.

**Q8**: Maybe the used inventories are biased by the scale of analysis and the adopted recognition methodology (small movements disappear) or by the used criteria to consider a landslide (for example only landslides that cause injuries). This and other assumptions related to the inventories should be deeply discussed in Discussion section;

**R8**: Thank you for your comment. We believe that this comment is quite similar with Q4, both about the representative of landslide inventories. Hence we have combined the replies of these two comments. Please refer to (Pg 9 Ln 36- Pg 10 Ln 6) in the newly revised manuscript for details.

**Q9**: Even if only a visual evaluation is possible to do: how different are the obtained results when compared with other global landslide susceptibility maps (some of them referred in this work)? And with other national/regional maps (for example, USA or Europe (Join Research Center))?

**R9**: Thank you for your comment. We have added relevant comparison as follows. It can be found in Pg 8 Ln 24-Pg 9 Ln 7.

The global landslide susceptibility map may be evaluated by comparison with four studies from the current literature that focus on large-scale landslide susceptibility.

Comparing the European landslide susceptibility map drawn by Van Den Eeckhaut et al. (2012) with the European part of susceptibility map in this study (Fig. 6 (a)), similar areas of high landslide susceptibility can be observed. The former map includes two levels (denoted High and Very High) as high susceptibility with a landslide probability of over 0.8, and this study also includes two levels (Levels 4 and 5) as high susceptibility with a probability over 0.7. The two maps have similar high susceptibility areas. Thus, for Europe, landslide susceptibility map in this study agrees with existing related study.

**Fig. 6** Comparison of existing studies with the related parts of this study

[Figure]

(a) Comparison of European landslide susceptibility map (from Van Den Eeckhaut et al. 2012) with the related part in this study's map

(b) Comparison of China landslide susceptibility map (from Liu et al. 2013) with the related part in this study's map

Comparing the Chinese landslide susceptibility map drawn by Liu et al. (2013)[1] with the China part of susceptibility map in this study (Fig. 6 (b)), the former map includes two levels (Levels 4 and 5) as susceptible with a landslide probability of over 0.6. Map in this study includes three levels (denoted Levels 3, 4 and 5) as susceptible with landslide probability of over 0.6. The main differences between the two maps are in the western Sichuan Basin and southern Tibet, which is famous for its high elevation and intense relative relief. This study applies many landslide cases in these areas. However, in the landslide database of Liu et al. (2013), only a few landslides occur in these areas. This discrepancy is the reason for the differences between the two maps.

As for landslide susceptibility at global scale, Nadim et al. (2006) and Hong et al. (2007) have ever made magnificent efforts on such topic. One global landslide susceptibility map (please refer to Fig. 7 in Nadim et al. (2006)) has five levels (Levels 5, 6, 7, 8 and 9) as susceptible, while the map from this study includes three levels (Levels 3, 4 and 5) as

[1] Liu, C., Li, W., Wu, H., Lu, P., Sang, K., Sun, W., & Li, R. (2013). Susceptibility evaluation and mapping of China's landslides based on multi-source data. Natural hazards, 69(3), 1477-1495.

susceptible. In general, the susceptible areas of these two maps are fairly similar except in Madagascar and the eastern Indo-China Peninsula.

Another global landslide susceptibility map (please refer to Fig. 3(a) in Hong et al. (2007)) has two levels (Levels 4 and 5) as susceptible, compared to map in this study, which has three levels (Levels 3, 4 and 5) as susceptible. These two maps are similar over Asia, Europe and Africa. However, it is noted that map of Hong et al. (2007) also differs from map of this study in that it shows high landslide susceptibility in central and southern India, and low landslide susceptibility in equatorial islands such as Malaysia, Indonesia, and the Philippines. We believe that the classification of landslide susceptibility of this research can be more scientific and closer to the existing conditions.

**Q10**: Page 3 line 33; Page 7 line 26 – please confirm the use of the term "topology". Topography? Please check the way how you performed in-text citations to several references: sometimes they are alphabetically (e.g. page 2 line 5), others chronologically (e.g. page 2 line 21) and others none of them (page 2 lines 17-18; page 3 line 25; page 4lines 15-16; page 5 lines 39-40; page 6 lines 14-15). Figures are adequate but in general with low resolution.
**R10**: Thank you for your comment. We have revised those parts mentioned above. In the new manuscript, we will provide figures with higher resolution. You can also find them in the attachment of this response.

**Q11**: Figure 1 (in figure or caption) should include time-period of the inventory and a reference to the main sources of the inventory. Each dot (triangle) means 1 landslide or more? Please provide a similar graphical scale in figures 1 and 2. I suggest 0-2000 km.
**R11**: Thank you for your comment. The time period and data reference have been added in Figure 1. Each dot represents one landslide. Similar graphical scale in Figures 1 and 5 (we believe that you refer to the table of global landslide susceptibility map produced by this research, i.e. Fig. 4 not Fig. 2 in the old version, Fig. 5 in the new version) has also been added. Please check the Figure 1 & 5 document in the attachment of this response.

**Response to Anonymous Referee #2**

In this document, the underlined part is those revision we made for a new manuscript.

**Question 1 (Q1)**: By considering landslide triggering factors, this work is more like a qualitative hazard mapping rather than a susceptibility mapping (van Westen, et al., 2008; Nadim et al., 2008; Fell et al., 2008). van Westen, C. J., et al. (2008). "Spatial data for landslide susceptibility, hazard, and vulnerability assessment: An overview." Engineering Geology 102(3-4): 112-131. Nadim, F., et al. (2006). "Global landslide and avalanche hotspots." Landslides 3(2): 159-173. Fell, R., et al. (2008). "Guidelines for landslide susceptibility, hazard and risk zoning for land use planning." Engineering Geology 102(3–4): 85-98.

**Response 1 (R1)**: Thank you for your comment. In the paper of van Westen et al. (2008), hazard assessment should include temporal and spatial probability of initiation, magnitude–frequency relation and run out potential. Because we did not study the temporal aspect of landslides, we hence did not use qualitative hazard mapping in this paper. On the other, when investigating landslide models, there may not be explicit distinctions between using the terms of susceptibility and hazard. For instance, in the paper of Daneshvar (2014)[2], the intensity of precipitation is considered as a critical factor when studying susceptibility, which influences the occurrence of landslides in semi-arid regions despite the small interval between the minimum and maximum precipitation. Thank you for your comment. We agree that our work is like a qualitative hazard mapping, but it is also acceptable to use susceptibility mapping.

**Q2**: The slope gradient factor should be added, which is as important as relative relief. Because, it is a common sense that steeper slopes are easier to have landslides than gentler ones.

**R2**: Thank you for your comment. We agree that slope is very important factor in the research of landslide susceptibility. We have included this factor when building landslide model. But the result show that it is not statistically significant. Therefore we did not include it in this paper. We have analysed the reason. At a global scale, factors such as elevation and slope gradient can be replaced by topographic index or relative relief, which indicate macroscopic differences in topography. Especially for landslide data with low location precision, using factors such as elevation or slope gradient that precisely relate to landslide location will reduce the accuracy of landslide susceptibility analysis. (Pg 3 Ln 42-Pg 4 Ln 2) This part has been added in the new manuscript.

**Q3**: In addition, land cover is also an important influencing factor on landslide susceptibility mapping. It is well acknowledged that vegetation, especially trees can prevent some shallow landslides. The authors are suggested to consider land cover types in their mapping.

**R3**: Thank you for your comment. Like the factor of slope, we included the factor of land cover when performing experiments. The land cover product with spatial resolution of 30m,
* * *
[2] Daneshvar, M. R. M. (2014). Landslide susceptibility zonation using analytical hierarchy process and gis for the bojnurd region, northeast of iran. Landslides, 11(6), 1079-1091.

GlobeLand30[3], is produced by scientists in China and submitted to United Nations for public use[4]. We tried this factor but found that it is not statistically significant and does not improve the model accuracy. Hence the factor of land cover is not included in this paper. The reason can be that comparing with other factors, land cover may not be a significantly important factor in assessing landslide susceptibility in global scale.

**Q4**: The authors used two datasets for dependent variable. Is there any consistency between them? Or, can you simply use them by combining both data sets? For example, maybe the Chinese datasets has more landslides within China while underestimate landslides abroad. Also, please introduce this new dataset in more detail, as there seems to be rare reports of it before.

**R4**: Thank you for your comment. The landslide data in this research comes from BNU World Geological Hazard Inventory and NASA global landslide inventory. In the former manuscript, we did not provide adequate information about it. We have added more in this manuscript:

(1)The sources of related databases. The items in World Geological Hazard Inventory were collected manually from news reports (e.g. mass media in China, Xinhua News, and Sina News) and records in books and journals. We searched information about landslide on Internet by using keywords like landslide and debris flow. Then we read these descriptions carefully to determine whether it is a landslide and locate it, and later put it into the database. Thus the main source of World Geological Hazard Inventory can be news data. By investigating these news, we can find out those landslides that are of large volume or of high danger, for these kinds of landslides can be of high news value. (Pg 6 Ln 16-23)

The NASA global landslide inventory mainly collects landslides from several existing databases, including International Consortium on Landslides website (ICL; http://iclhq.org); International Landslide Centre, University of Durham (ILC; http://www.landslidecentre.org); The EM-DAT International Disaster Database (http://www.em-dat.net); International Federation of Red Cross and Red Crescent Societies field reports (http://www.ifrc.org); Reliefweb (http://reliefweb.int); humanitarian disaster information run by the United Nations Office for the Coordination of Humanitarian Affairs (OCHA); other online regional and national newspaper articles and media sources.(Pg 6 Ln 8-15)

The best resolution of World Geological Hazard Inventory is 0.001 degree, about 100m. (Pg 6 Ln 27) The NASA global landslide inventory 2km. (Pg 6 Ln 15-16)

(2) The time period of landslide database. In the World Geological Hazard Inventory, the earliest event can be dated to 1618. In this database, there is 117 landslides occurred before 1975, 84 between 1975 to 2000, and 274 between 2000 and 2014. The landslide events in the NASA global landslide inventory mainly happened in 2003, 2007, 2008 and 2009. Hence these two databases are complementary and they can be emerged to produce a more complete landslide database.(Pg 6 Ln 36-40) This part has been added in the new manuscript.

(3) The combination of two databases. When combining these two databases, the occurrence of time provides crucial standard. When two landslide events have different
* * *
[3] http://www.globeland30.org/GLC30Download/index.aspx
[4] https://unstats.un.org/unsd/GlobeLand30.htm time (month), they are both reserved in the new database. If two events have the same
occurrence time (month) and their locations are close, investigation through details in
source could determine whether they are from the same disaster. If yes, the record with
higher spatial resolution is reserved and the one with lower resolution is dropped. (Pg 6 Ln
30-34) This part has been added in the new manuscript.
**Q5**: An improved discussion is needed to compare and highlight the contribution of this
work in global landslide mapping compared to previous works.
**R5**: Thank you for your comment. We have added relevant comparison as follows. It can
be found in Pg 8 Ln 24-Pg 9 Ln 7.

[revised manuscript text omitted]

[Figure]
**Fig.**3 4 ROC curve of validation process

[Figure]

**Fig.4 5** Global-scale landslide susceptibility map

[Figure]

批注 [LL24]: Reviewer #1, R10, R11

**Fig. 6** Comparison of existing studies with the related parts of this study

[Figure]

(a) Comparison of European landslide susceptibility map (from Van Den Eeckhaut et al. 2012) with the related part in this study's map

(b) Comparison of China landslide susceptibility map (from Liu et al. 2013) with the related part in this study's map

---

## Author Response (AR2)

**Response to comments**

The sentences underlined are the added parts in the newly revised manuscript. The numbers of pages and lines are used to locate related parts in the tracked revised manuscript.

**Report #1**

By Anonymous Referee #2

Q1 (Question 1): The Discussion still needs improvement. The first paragraph of the Discussion section is only one sentence, and it is too short to be a standing paragraph. In addition, solely comparing with others' finding is not applicable. Also the language of the paper should be improved by a native speaker or language editing services.

R1 (Reply 1): Thank you for your comment. We have rewritten the first paragraph of the Discussion section to make it better. To evaluate the accuracy of susceptibility map produced in the research, the global landslide susceptibility map is compared with four studies from the current literature that focus on large-scale landslide susceptibility. In regional scale, two landslide susceptibility maps, i.e. European (Van Den Eeckhaut et al. 2012) and Chinese (Liu et al. 2013), are selected. In global scale, the studies of Nadim et al. (2006) and Hong et al. (2007) are selected. (Page 8 Line 25-29) We have tried to improve the language of this paper. But because we are not provided with enough time for language editing, the manuscript we submitted this time have not been improved in language yet. Once the language editing is finished, we will update the manuscript immediately.

 **Report #2**

By Referee #1: Ricardo A.C. Garcia, rgarcia@campus.ul.pt

Q1: Fig 2 – Please improve figure caption. Insert sources and indication of which map is/is not produced by the authors or by others. Is it possible to have a general % of agreement?

R1: Thank you for your comment. We have improve the figure caption of Fig. 2. It has been changed to: Comparison of landslide overlay in Europe. (Page 22) In the figure, we also added the sources of the maps. In addition, we provide a general estimation of the agreement. It is estimated that there is about 60% agreement between these two landslide distributions in general. (Page 7 Line 5-6)

Q2: Pg. 7 line 20-21 – Please refer that the subset inventories (model/validation) are proportional to the weight of each continent data set in the global inventory. Is this idea true?

R2: Thank you for your comment. In this paper, the total database which consists of landslides and non-landslides is split into two subsets, one for modelling and the other for validation. According to Van Den Eeckhaut et al. (2012), the percentage for this split can be 70%:30%. Then we applied it to this research. It means that the landslide inventory for modelling has 70% of total landslides and 70% of total non-landslides and the landslides and non-landslides are randomly selected from the total database, during which the attribute of continent is not considered. It is clear that this splitting percentage does not have anything to do with the landslide data in each continent.

Q3: References: Van Den Eeckhaut et al. - Statistical modelling of Europe-wide landslide susceptibility using limited landslide inventory data. Landslides (2012) 9:357–369 DOI

10.1007/s10346-011-0299-z is missing

R3: Thank you for your comment. After checking the reference list, we find that this reference exists already. Please refer to (Page 14 Line 4-6).

[revised manuscript text omitted]